# Microglia sense fungal infections through capsular components from capillary-bound *Cryptococcus neoformans* via endothelial nucleotide signaling

Chenxu Feng[1☯], Ge Wang[1,2☯], Yixuan Wang[1☯], Xiang Gao[1], Zhenqi Xu[1,2], Luyao Fang[2], Ziyi Ma[1], Suwei Zheng[1], Yuyan Xie[3], Yufeng Chu[1¤a], Mei Meng[1¤b], Angela Yang[1], Miriam Lu[1], Judd Denzel Garcia Mondina[1], Weiwei Zhu[1], Lisheng Zhang[2], Linqi Wang[3], Zongyan Chen[2]*, Donglei Sun[1,4]*

**1** Sheng Yushou Center of Cell Biology and Immunology, School of Life Sciences and Biotechnology, Shanghai Jiao Tong University, Shanghai, China, **2** Shanghai Veterinary Research Institute, Chinese Academy of Agricultural Sciences, Shanghai, China, **3** State Key Laboratory of Mycology, Institute of Microbiology, Chinese Academy of Sciences, Beijing, China, **4** Shanghai Key Laboratory for Antibody-Drug Conjugates with Innovative Target, Shanghai, China

☯ These authors contributed equally to this work.
¤a Current Address: Department of Neurology, Shandong Provincial Hospital Affiliated to Shandong First Medical University, Jinan, China
¤b Current Address: Department of Critical Care Medicine, Shanghai Ninth People's Hospital, Shanghai Jiao Tong University School of Medicine, Shanghai, China
* zychen@shvri.ac.cn (ZC); dongleisun@sjtu.edu.cn (DS)

## Abstract

Macrophages are essential for host defense, yet how parenchyma-residing macrophages detect pathogens without direct contact remains unclear. *Cryptococcus neoformans* is an encapsulated fungal pathogen that infects the brain. Using in situ imaging of mouse model, we showed that brain-resident microglia vigilantly detect capillary-residing *C. neoformans* prior to its blood–brain barrier transmigration, but are less responsive to nonencapsulated fungi or parenchyma-injected *C. neoformans*. Microglia migrate to and enwrap leaky capillaries harboring fungi, leading to fungal uptake but not clearance, instead promoting fungal growth. Microglial response is triggered by released capsule components, rather than the assembled capsule. In particular, glucuronoxylomannan (GXM) plays a critical role by activating endothelial cells to release nucleotides which act on microglia P2Y12. Our findings revealed a novel paradigm by which microglia detect pathogens without direct contact, offering new insights for microglia-directed antifungal therapies.

## Introduction

Macrophages exist widely across various organs, playing crucial roles in homeostasis and host defense [1]. While macrophages located within blood vessels and airways, such as Kupffer cells [2] and alveolar macrophages [3], can directly interact with

**Data availability statement:** All relevant data are within the paper and its Supporting information files. All numerical data are found in S1 Data, which contains multiple datasheets. FCS files are available on Zenodo (https://doi.org/10.5281/zenodo.18398298).

**Funding:** This work was supported by the National Key Research & Development Plan of China (2023YFC2306300), National Natural Science Foundation of China (32100723) to DS; Natural Science Foundation of Shanghai(24ZR1479200), Chinese Academy Of Agricultural Sciences Agricultural Science and Technology Innovation Program (CAAS-ASTIP-2026-SHVRI) to ZC; Shanghai Oriental Talents Plan (T2024107) to LZ. The funders had no role in study design, data collection and analysis, decision to publish, or preparation of the manuscript. No author received salary from any of the funders.

**Competing interests:** The authors have declared that no competing interests exist.

**Abbreviations:** BBB, blood–brain barrier; CFUs, colony-forming units; DT, diphtheria toxin; GXM, glucuronoxylomannan; i.n., intranasally; PBS, phosphate-buffered saline; TRITC, tetramethylrhodamine isothiocyanate.

invading pathogens, many parenchyma-residing macrophages lack direct access to the bloodstream or mucosal interfaces. In the brain, microglia are the predominant parenchyma-residing macrophage constantly surveying the brain for dying cells [4] and invading pathogens [5]. As brain resident macrophages, microglia are essential during development and critically involved in various conditions, including focal injury [6], neurodegenerative diseases [7], stroke [8], etc. Moreover, microglia actively protect the host against pathogens that have crossed the blood–brain barrier (BBB) and enter the parenchyma [9,10]. However, little is known on whether and how microglia detect pathogens prior to their BBB transmigration.

*Cryptococcus neoformans* is a typical fungal pathogen that is capable of BBB transmigration via vasculature [11]. This encapsulated fungus causes fatal cryptococcal meningoencephalitis, claiming over 110,000 lives annually [12], and has been listed as the top-priory fungal pathogen by World Health Organization. Although the infection initiates from the respiratory system, *C. neoformans* can escape the lung and disseminate to the brain via circulation. Due to its relatively large size and incapability to deform within microvessels, *C. neoformans* tend to be captured in brain capillaries following dissemination [11]. As a result, BBB transmigration by *C. neoformans* is believed to occur primarily at the capillary level [13]. Earlier histological studies observed that host microglial cells cluster around the *C. neoformans* colonies in infected brain sections [14]. Subsequent investigations confirmed that microglia can interact with *C. neoformans* following BBB invasion [15,16], but failed to control fungal proliferation [15,17]. In a zebrafish model, microglia were shown to actively transfer yeast cells across the BBB or phagocytize them immediately after crossing [18]. These studies highlighted the potentially active responses of microglia during *C. neoformans* pathogenesis, however, the timing of microglial responses towards *C. neoformans* and the underlying mechanisms remain poorly understood.

*C. neoformans* expresses various virulence factors, among which the polysaccharide capsule is the most well-characterized. The cryptococcal capsule is primarily composed of glucuronoxylomannan (GXM, ~90%) and galactoxylomannan (GalXM, 5%–8%) [19], which are thought to be synthesized in the endoplasmic reticulum and Golgi apparatus, followed by vesicle-mediated secretion [20–22]. GXM is a high-molecular-weight polymer composed of an α-1,3-linked mannan backbone with glucuronic acid and xylose side chains, whereas GalXM is an α-1,6-galactan backbone with galactomannan side chains decorated with xylose residues. Over the past decades, a series of capsule-associated genes (CAP genes) have been identified; disruption of which typically results in an acapsular phenotype and impaired virulence [19,23]. However, the exact functions of these CAP genes remain poorly characterized. For example, *cap59* and *cap60* share homology with mannosyltransferase, and *cap59* additionally contribute to extracellular trafficking of GXM [20]. *cap10* shares homology with a xylosyltransferase, while *cap64* is likely involved in GXM O-acetylation [24]. In addition to CAP genes, other genes such as *uge1* (encoding UDP-glucose epimerase [25]) and *uxs1* (encoding UDP-xylose synthase [26]) also play critical roles in shaping capsule structure. Despite this knowledge, how individual capsule components influence *C. neoformans* interaction with microglia remains largely unexplored.

Fluorescence imaging provides a powerful tool for studying host-pathogen interactions. Using intravital microscopy, we previously showed liver-resident macrophages capture circulating *C. neoformans* [27] and that fungal infection triggers monocyte recruitment to brain vasculature [28]. During previous brain intravital imaging [28], we observed frequent recruitment of microglia towards *C. neoformans* trapped in capillaries. In this study, using in situ imaging, we further characterized microglia recruitment and their underlying mechanisms in detail. By showing microglia vigilantly detect capillary-residing *C. neoformans* prior to its BBB transmigration, current study demonstrated a paradigm for parenchyma macrophage to sense and respond to pathogens without direct contact.

## Results

### Microglia efficiently respond to *C. neoformans* captured in the brain

The brain is the primary target for *C. neoformans*, which harbors a dense population of resident macrophages known as microglia. To study the responses of microglia during fungal brain invasion, we intranasally infected CX3CR1$^{gfp}$ reporter mice (microglia labeled) [29] with tdTomato-expressing *C. neoformans* H99 strain (H99-tdT, serotype A) as a natural infection model. Mouse brains were collected for in situ imaging of cortex region by whole-mount confocal approach to preserve the native interaction information (Fig 1A). *C. neoformans* forms colonies of varying sizes in the brain, which are surrounded by abundant GFP$^+$ cells (Figs 1B and S1A), indicating microglial responsiveness to fungal invasion. The colony periphery exhibited intensive accumulation of GFP$^+$ cells, while the inner regions contained substantial GFP$^+$ cell debris (Fig 1B). Since monocytes also express GFP in CX3CR1$^{gfp}$ mice, we used CX3CR1$^{gfp}$CCR2$^{rfp}$ dual-reporter mice to characterize the contribution of monocytes in this model. Although CCR2$^+$ monocytes were recruited to the fungal colonies, cells interacting with *C. neoformans* were predominantly CCR2$^-$, confirming their microglia identity (S1B Fig). Time course study revealed sustained microglia association as colonies expanded (Fig 1C and 1D).

To overcome the variability of intranasally (i.n.) infection and investigate how early microglia can respond to *C. neoformans* captured in the brain, we switched to i.v. infection model, in which fungal cells are rapidly trapped in brain capillaries after injection (S1C Fig) [11,28,30]. Remarkably, the majority of *C. neoformans* cells were already associated with microglia 24 h post-infection (Fig 1E and S1 Movie). Higher-resolution imaging revealed close physical interactions between microglia and the fungi (S2 Movie). Time-course analysis further showed microglial responses begin as early as 6 h post infection (Fig 1F). We further demonstrated that *C. neoformans* can proliferate in the presence of microglia association (Fig 1G and 1H). Monocyte depletion by Clodronate liposome did not reduce microglial association with *C. neoformans* (S1D Fig), confirming that the responding microglia are not monocyte-derived. Following recruitment towards *C. neoformans*, microglia exhibited enhanced GFP expression, making them conspicuous among resting microglia (S1E Fig). These microglia also displayed pronounced morphological changes, represented by increased cell body and retracted dendrites (S1F–S1H Fig), hallmarks of microglia activation [31]. Interestingly, although microglia in the cerebellum typically exhibit longer dendrites than those in the cortex (S1I Fig), they are equally responsive to fungal invasion (S1J Fig).

Notably, high dose fungal infection through the tail vein led to rapid death of the host (S1K Fig). To rule out side effects of high dose i.v. infection of *C. neoformans*, we also confirmed similar microglial response using a much lower dosage (S2A and S2B Fig). In addition to high-virulent serotype A strain, microglia also respond to serotype D strain with equivalent efficiency (S2C Fig). *C. neoformans* enlarges its body size within the host environment. Notably, we found the few fungal cells lacking microglial association often appeared smaller, likely due to the lack of viability (S2D Fig). To investigate whether fungal viability influence microglial response, we used heat-killed fungi and confirmed fungal viability is essential for microglial response (S2E Fig). Similarly, inert polystyrene beads of comparable size did not trigger microglia recruitment (S2F Fig). Importantly, microglial recruitment does not require fungal proliferation, as the *ras1Δ* mutant, defective in growth at 37 °C [32] (S2G Fig), induced microglia recruitment to a similar level as the wild-type strain (S2H Fig). We also noted that, although microglia responded actively towards *C. neoformans*, they did not show much increase

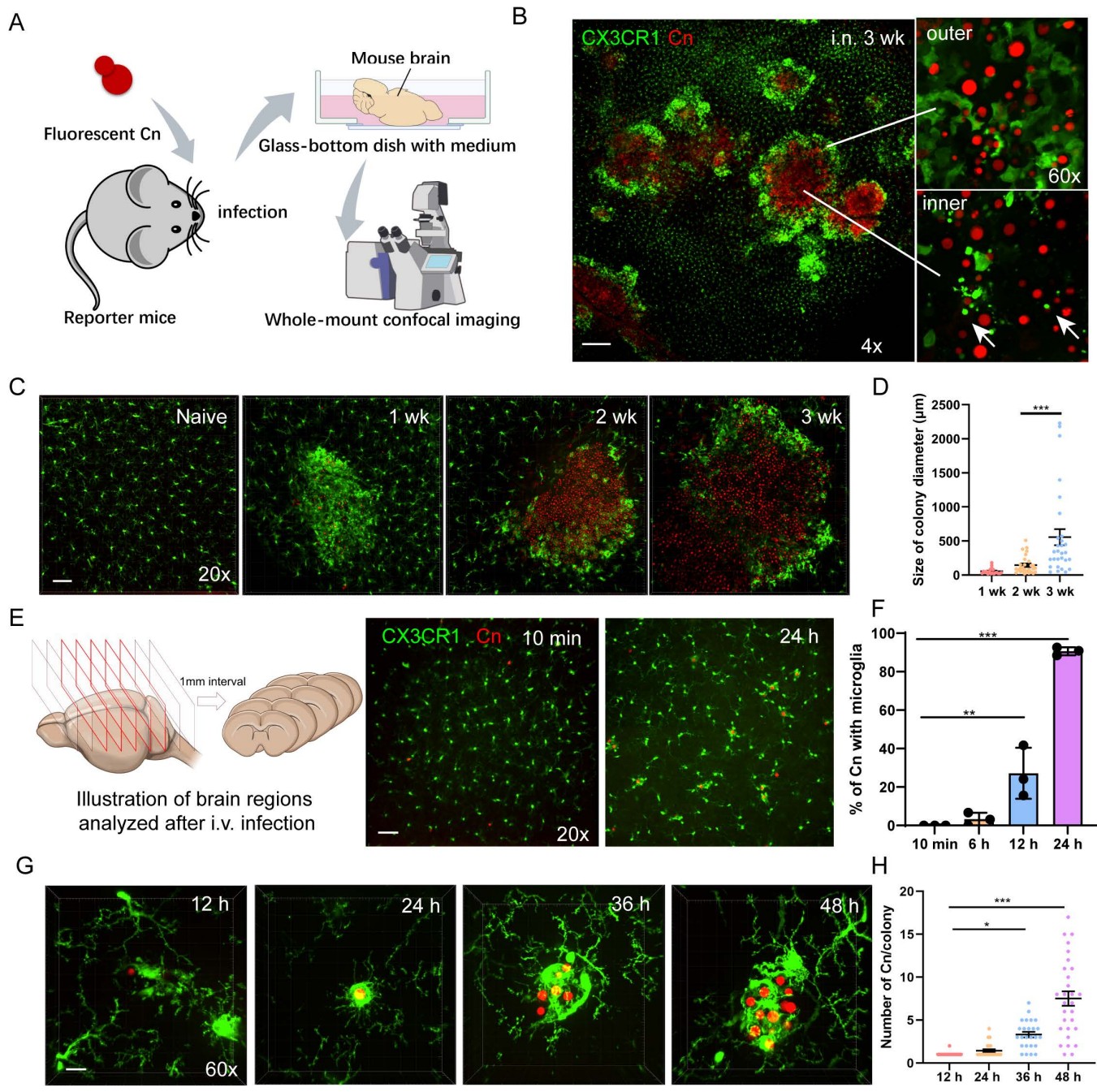

**Fig 1. Microglia rapidly respond to brain invasion by *Cryptococcus neoformans*. (A)** The illustration of in situ imaging procedure. **(B)** Representative images showing the response of CX3CR1+ microglia towards *C. neoformans*. CX3CR1gfp/+ mice were i.n. infected with $1 \times 10^4$ tdTomato-labeled *C. neoformans* H99 strain (H99-tdT) for 3 weeks. The brains were imaged after tissue clearing. Arrows indicate cell debris from GFP+ cells. **(C)** The association of microglia with *C. neoformans* in the brain at various time points after i.n. infection. **(D)** The size distribution of fungal colonies in the brain after i.n. infection at various time points. Each dot represents an individual fungal colony pooled from five mice per group. **(E)** CX3CR1gfp/+ mice were i.v. infected with $1 \times 10^7$ H99-tdT for indicated time. The brains were sliced into 1 mm coronal sections, and the 3th–7th slice from anterior to posterior were imaged after tissue clearing for quantification (Left). Representative images showing the localization of microglia with *C. neoformans* after i.v. infection at indicated time points (Right). **(F)** The percentage of *C. neoformans* association with microglia at indicated time points. CX3CR1gfp/+ mice ($n = 3$ mice/group) were i.v. infected with $1 \times 10^7$ H99-tdT for different time points. **(G)** Higher magnification 3D images showing the association of microglia with *C. neoformans* in the brain after i.v. infection at indicated time points. **(H)** The number of fungal cells per colony in the brain following i.v. infection at indicated time points. The data underlying this Figure can be found in S1 Data. Scale bar: 200 μm (B), 50 μm (C and E), 20 μm (G). ** $p < 0.01$, *** $p < 0.001$ by one-way ANOVA (D) (F) (H).

in activation markers like CD11c and MHCii in this acute infection model (S2I Fig). Collectively, *C. neoformans* captured in the brain elicits rapid microglial responses, which depends on fungal viability but not proliferation.

## Microglia respond to *C. neoformans* prior to its BBB transmigration

Since *C. neoformans* reaches the brain via circulation, we next investigated the influences to brain vasculature during microglia-fungi interaction. Under steady conditions, microglia are homogenously distributed in the brain parenchyma, with most cells located at a distance from the capillary vessels and a small fraction located juxtavascularly (Fig 2A and S3 Movie). *C. neoformans* brain infection results in intimate association of microglia with the fungi-residing loci of the capillaries (Fig 2B). Although *C. neoformans* is well known for its ability to cross the BBB [13,33,34]; surprisingly, we found almost all microglial responses occurred prior to fungal transmigration to the abluminal side of the BBB (Fig 2C). Direct intracerebral injection of *C. neoformans* into the brain parenchyma triggered limited microglial responses compared to fungi emerging from the vasculature (S3A Fig), emphasizing the importance of the vasculature origin for fungal induced microglial responses. Further labeling revealed that microglial responses were directed almost exclusively toward mother cells rather than daughter cells (Fig 2D). In rare cases where mother cells transmigrated, microglial association with daughter cells was also detected (S3B Fig and S4 Movie).

   Microglia actively migrated to and enwrapped the fungi-residing loci of the capillaries, ultimately resulted in fungal phagocytosis (Fig 2E and S5 Movie), likely associated with endothelial cell retraction. The chemokine receptor CX3CR1 has been implicated in the microglia recruitment in other models [31]. However, we found the CX3CR1$^{gfp/gfp}$ mice, lacking functional CX3CR1, exhibited comparable microglial responses to *C. neoformans* (S3C and S3D Fig). We next studied the association of *C. neoformans* with immune cells by flowcytometry. Interestingly, microglia, defined as CD45$^{int}$CD11b$^+$ (S3E Fig), associated with more fungi than other leukocytes (S3F and S3G Fig). Interestingly, CX3CR1 deficiency did resulted in a modest reduction in fungal association by microglia (S3F and S3G Fig), likely due to reduced phagocytosis.

   *C. neoformans* is known to compromise BBB integrity [13]. Consistently, the capillary-enwrapping microglia often contained inclusion bodies that incorporated the vessel-labeling dye, indicative of local vascular leakage (Fig 2F). After analysis of the leakage around capillary loci with or without *C. neoformans*, we found that microglia recruitment is strongly associated with vascular leakage (Fig 2G and 2H). Collectively, microglia respond to *C. neoformans* by migration to and enwrapping the capillaries prior to fungal BBB transmigration, a process associated with capillary leakage.

   Interestingly, CX3CR1$^+$ monocytes were frequently observed in close proximity to *C. neoformans* in the capillary (S3H and S3I Fig). However, unlike microglia, monocytes were largely unable to engulf *C. neoformans* efficiently from the capillary lumen (S3J Fig) and were frequently positioned on one side of the fungi (S3H Fig), consistent with flowcytometry results, possibly due to the complete blockade of capillaries by the fungi. Collectively, microglia respond to *C. neoformans* prior to its BBB transmigration.

## Nonencapsulated fungi fail to elicit robust microglial responses

Intrigued by the robust sensitivity of microglia in detecting *C. neoformans*, we next questioned whether microglia can detect other nonencapsulated fungal species or not. We found that *Saccharomyces cerevisiae* failed to induce microglia recruitment in the brain, even after prolonged infection (Fig 3A and 3B). Interestingly, *Candida albicans*, another common fungal pathogen, grows hyphae in the brain (Fig 3C and 3D). Although hyphae growth of *C. albicans* can penetrate endothelial layers of the brain (Fig 3E), it did not induce substantial microglia recruitment as *C. neoformans* (Fig 3F and 3G). Thus, the high efficiency of microglial response is unique to the encapsulated *C. neoformans*, but not to nonencapsulated fungal species.

## Microglia association promotes fungal proliferation

To study the consequences of microglia recruitment towards *C. neoformans*, microglia were depleted using the CSF1R inhibitor PLX3397 mixed in rodent diet. Treatment of PLX3397 diet for 14 days resulted in >90% of microglia depletion

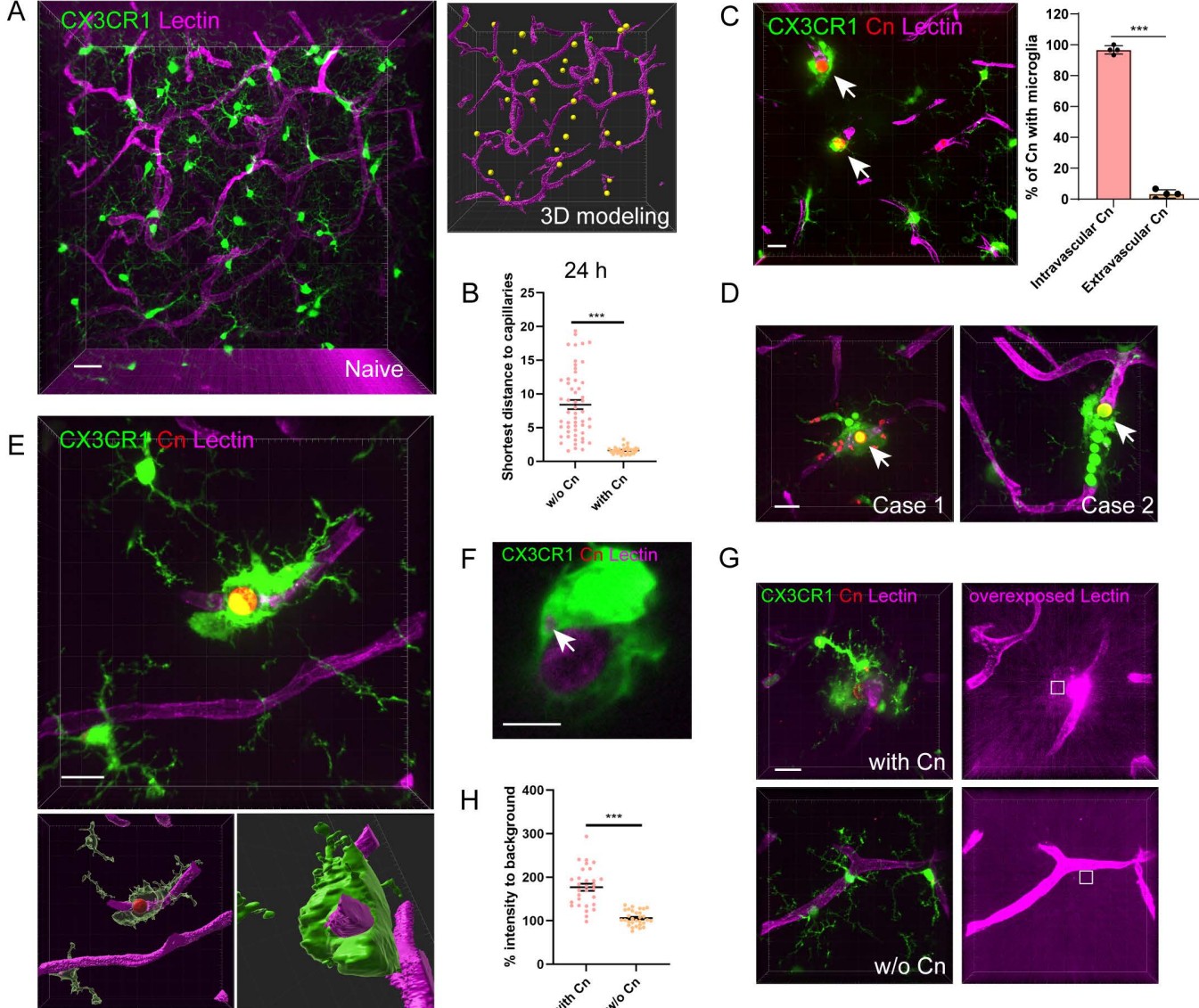

**Fig 2. Microglia respond to *Cryptococcus neoformans* prior to its BBB transmigration. (A)** The distribution of microglia in the cortex region of naïve CX3CR1$^{gfp/+}$ mouse brain (left) and 3D modeling showing the relative positioning of microglia and blood vessel (right). Microglia located <2 μm from nearest blood vessel were shown in green. The vasculature was labeled by Dylight-649 conjugated Tomato-Lectin. **(B)** The quantification of shortest distance to capillaries for microglia with or without *C. neoformans* association. CX3CR1$^{gfp/+}$ mice were i.v. infected with 1 × 10$^7$ H99-tdT for 24 h. Each dot represents an individual microglial cell pooled from three mice per group. **(C)** Representative image showing microglia association with *C. neoformans* prior to fungal BBB transmigration (left). Arrows indicate the fungal-residing loci. The percent location of *C. neoformans* (inside or outside capillaries) out of all fungi with microglia association was shown to the right. CX3CR1$^{gfp/+}$ mice (*n* = 4 mice/group) were i.v. infected with 1 × 10$^7$ H99-tdT for 24 h. **(D)** Representative images showing microglia tend to associate with mother cells than daughter cells. Red indicates mother cells (arrow). **(E)** Representative image showing the enwrapping of blood vessels and engulfment of *C. neoformans* by microglia. 3D surface modeling of *C. neoformans*, microglia, and vasculature was shown below. **(F)** Representative image showing the phagocytosis of dye by microglia, suggesting vessel leakage. Since an overdose of lectin was used, unbound free lectin in the circulation may leak outside the capillaries. **(G)** Representative images showing the leakage of dye from the fungal-residing loci of capillary (upper) and control region without *C. neoformans* association (lower). Examples of sampling areas (5 μm x 5 μm) were shown as squares. **(H)** The quantification of dye leakage from capillaries with or without *C. neoformans*. The data underlying this Figure can be found in S1 Data. Scale bar: 20 μm (A) (C) (D) (G), 5 μm (F), 10 μm (E) (G). *** *p* < 0.001 by unpaired student *t* test (B) (C) (H).

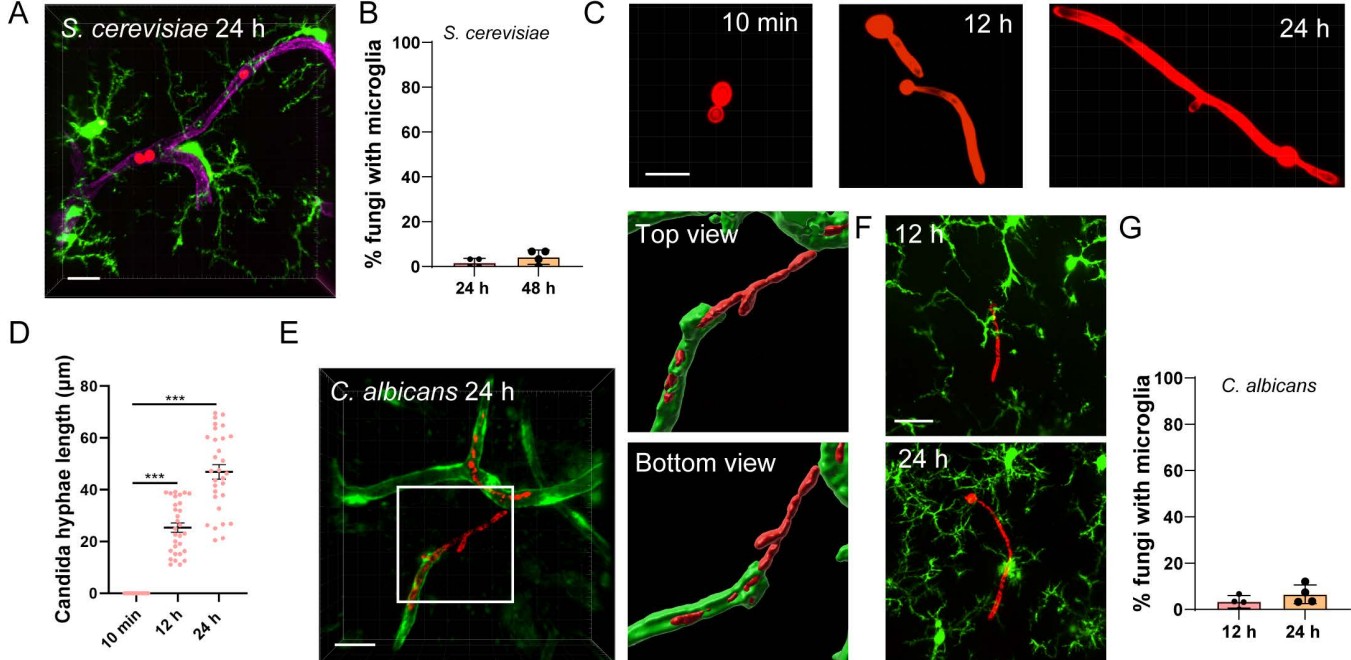

**Fig 3. Nonencapsulated fungi fail to elicit robust microglial responses. (A)** Representative image showing *Saccharomyces cerevisiae* cells (red) located inside brain capillary vessel without microglia association. CX3CR1$^{gfp/+}$ mice were i.v. infected with $1 \times 10^7$ TRITC labeled *S. cerevisiae* for 24 h. Blood vessels were labeled by Dylight 649-conjugated Tomato Lectin. **(B)** The percentage of *S. cerevisiae* with microglia association 24 and 48 h post infection. **(C)** Representative images showing the growth of hyphae by *Candida albicans* in the brain at indicated time points post-infection. mice were i.v. infected with $1 \times 10^7$ tdTomato-expressing *Candida albicans*. **(D)** The length of hyphae growth by *C. albicans* in the brain at indicated time points post-infection. Each dot represents an individual fungal colony pooled from three mice per group. **(E)** A representative imaging and 3D reconstructed images viewed from different angles showing the penetration of *C. albicans* across blood vessel (GFP from Tie2-GFP mice). **(F)** A representative image showing the lack of association between *C. albicans* and microglia at indicated time points. CX3CR1$^{gfp/+}$ mice were i.v. infected with $1 \times 10^7$ tdTomato-expressing *C. albicans* for 12 and 24 h. The association of microglia and fungi were calculated. **(G)** Quantification of microglia association with *C. albicans*. The data underlying this Figure can be found in S1 Data. Scale bar: 20 μm (A) (C) (E) (F). *** $p < 0.001$ by one-way ANOVA.

(Fig 4A and 4B). Interestingly, despite substantial microglia depletion, the remaining microglia remained responsive to *C. neoformans* (Fig 4C), although microglial association were less frequent (Fig 4D). Microglia depletion led to slightly smaller fungal colonies (Fig 4E), consistent with a reduction in brain fungal burden, but not in other organs (Fig 4F). Notably, PLX3397 may have profound systemic and cellular effects, especially its monocyte inhibitory roles. To independently validate these findings, we also employed a genetic microglia depletion model using CX3CR1$^{CreER}$iDTR mice. Tamoxifen-induced Cre recombinase expression followed by diphtheria toxin (DT) treatment effectively ablated microglia (Fig 4G), and this depletion also significantly reduced brain fungal burden (Fig 4H). These results confirmed that microglia promote *C. neoformans* growth in the brain [15].

### Fungal GXM is crucial for microglial response

Capsule production in *C. neoformans* is governed by several capsule-associated genes, among which, *cap59Δ*, *cap60Δ* and *cap64Δ* are the first identified. We confirmed the lack of capsule for these mutants in vivo (Fig 5A and 5B). Notably, all three mutants showed impaired microglia association (Fig 5C and 5D), with *cap59Δ* elicited the weakest response followed by *cap60Δ*, and *cap64Δ* showed the strongest response among them (Fig 5D and S6–S8 Movies). Heat-killed *C. neoformans* failed to trigger microglial response, suggesting capsule building blocks, not assembled capsule, mediate the response (S2E Fig). Since cryptococcal capsule consists of GXM and GalXM (Fig 5E), and the *cap59Δ* mutant is GXM

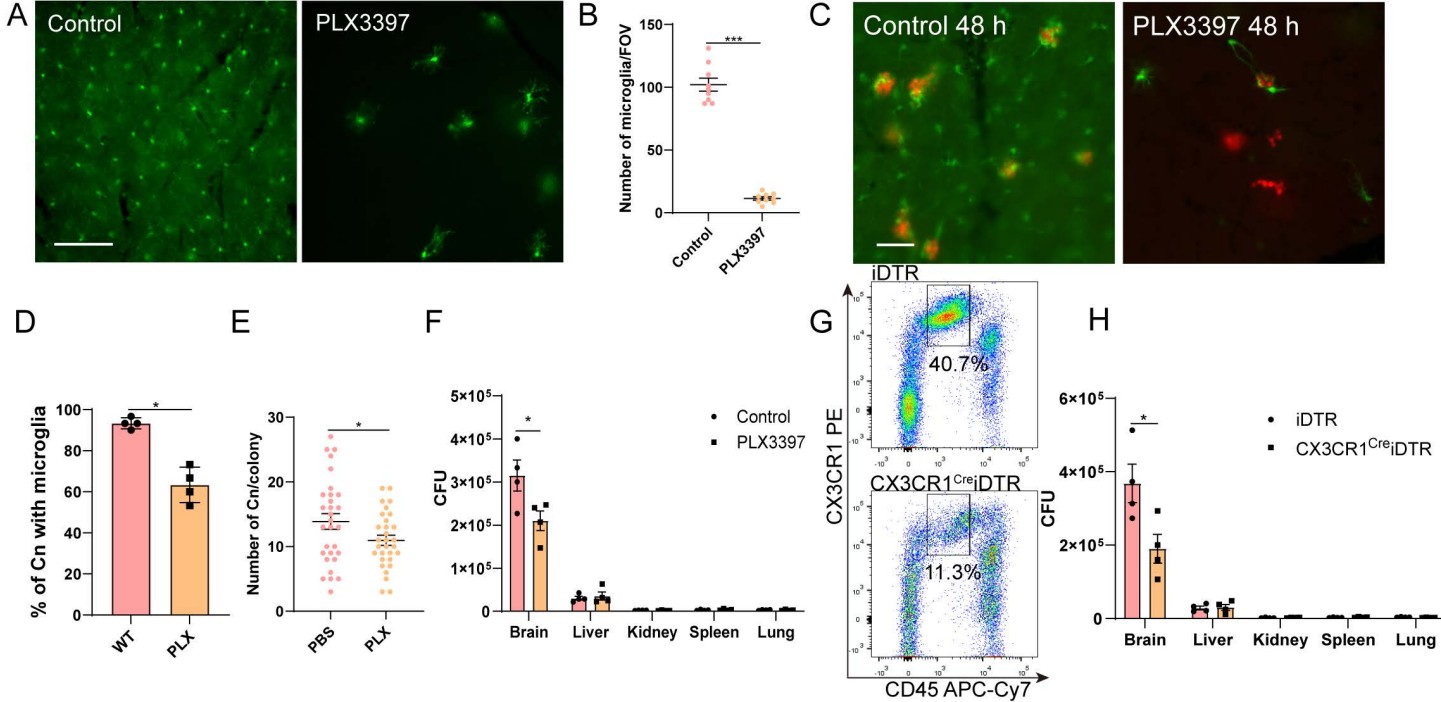

**Fig 4. Microglia promote _Cryptococcus neoformans_ proliferation in the brain. (A)** Representative images showing the successful depletion of brain microglial by feeding mice 2 weeks with a diet containing 1,200 mg/kg CSF1R inhibitor PLX3397. **(B)** Quantification of microglia per field of view (0.2 × 0.2 mm). **(C)** Representative images showing the association of microglia with _C. neoformans_ with or without treatment of PLX3397 after i.v. infection with $1 \times 10^7$ H99-tdT for 48 h. **(D)** The percentage of _C. neoformans_ with microglia association 48 h post-infection. **(E)** The number of fungal cells per colony with or without treatment with PLX3397. Each dot represents an individual fungal colony pooled from four mice per group. **(F)** The fungal burden in different organs after i.v. infection with a low dose ($5 \times 10^4$) _C. neoformans_ for 3 days with or without PLX3397 treatment. **(G)** Representative flowcytometry images showing the depletion of microglia (CD45$^{int}$ CX3CR1$^+$) after treating CX3CR1$^{CreER}$iDTR or control iDTR mice with diphtheria toxin (DT). Refer to S7C Fig for gating strategy. **(H)** The fungal burden in different organs after i.v. infection with a low dose $5 \times 10^4$ H99 strain for 3 days after treatment with DT. The data underlying this Figure can be found in S1 Data. Scale bar: 50 μm (A), 20 μm (C). * $p < 0.05$, unpaired student _t_ test (B) (D) (E) (F) (H).

negative but GalXM positive [25]; the minimal microglial response in _cap59Δ_ highlights GXM as the key component. Furthermore, _cap59Δ_ infection did not disrupt the BBB, as indicated by preserved basement membrane (S4A Fig) and intact tight junction protein ZO-1 (S4B Fig).

GalXM, the minor component of capsule, requires UDP-glucose epimerase (encoded by _uge1_) for production [25]. Accordingly, the _uge1Δ_ strain is GalXM deficient but GXM positive [25]. We found _uge1Δ_ strain triggered comparable microglial responses as wild-type strain (Fig 5F and S9 Movie), indicating GalXM is dispensable for microglia recruitment. Both GXM and GalXM contain xylose residues, whose synthesis requires UDP-xylose encoded by _uxs1_ gene [35]. Surprisingly, _uxs1Δ_ mutant also induced comparable microglial response as wild-type strain (Fig 5G), suggesting the UDP-xylose production is dispensable for microglia recruitment. Notably, the _uge1Δ_ and _uxs1Δ_ strains exhibited altered capsule thickness in vivo (S5A and S5B Fig), as previously reported [25].

To identify other virulence factors that contribute to microglial response, we performed a screening of virulence factors based on literature for their microglial responses (S1 and S2 Tables). The _pka1Δ_ mutant, encoding the catalytic subunit of cyclic adenosine 5′-monophosphate-dependent protein kinase A, was identified to have dramatically reduced microglia recruitment (S5C and S5D Fig; S10 Movie). This mutant displayed substantially reduced capsule thickness as previously reported (S5E and S5F Fig) [21]. We confirmed that _pka1Δ_ strain produces less GXM (S5E Fig), supporting GXM as the

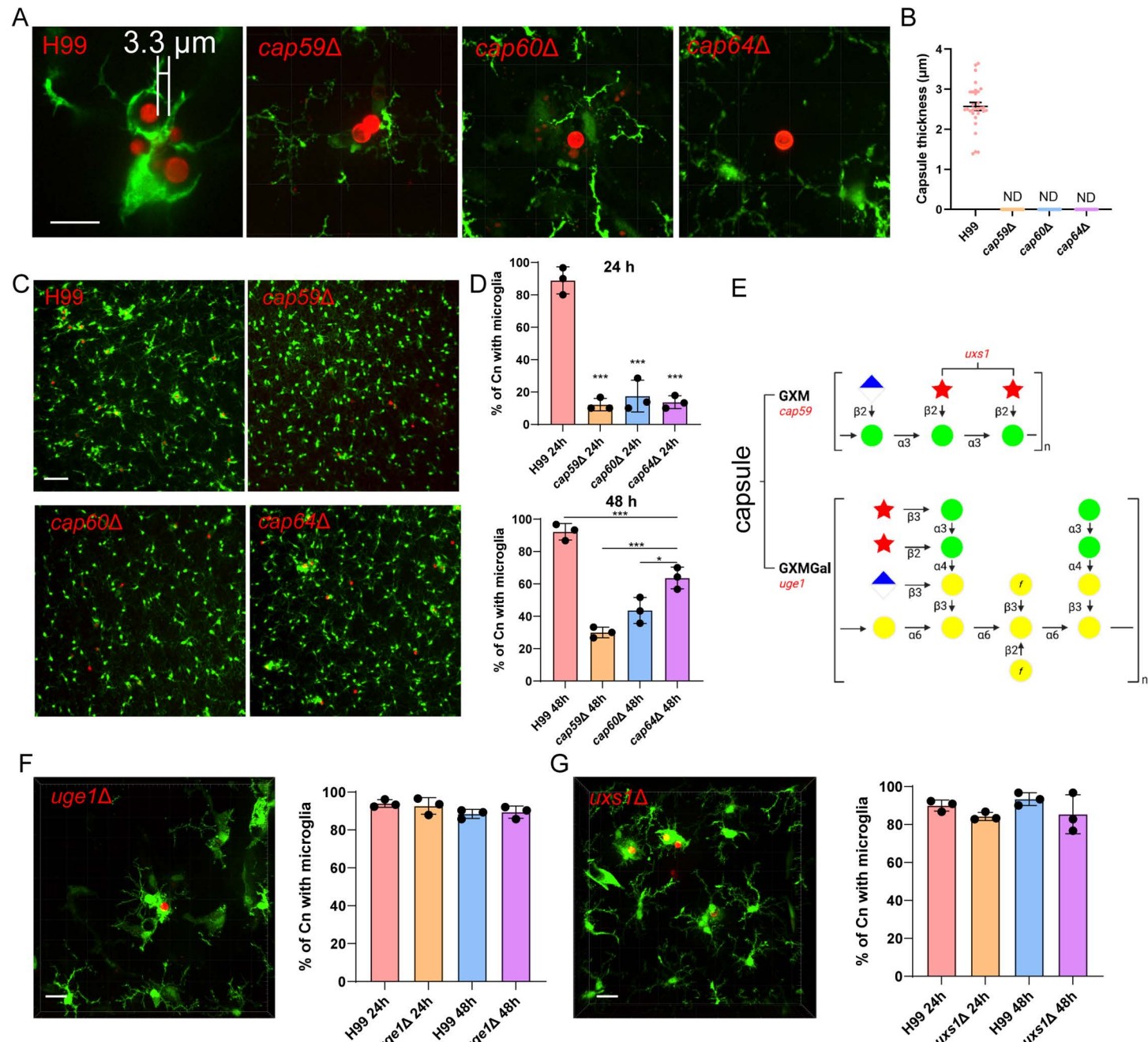

**Fig 5. Microglia association with *Cryptococcus neoformans* depends on capsule production.** (A) Representative images showing the capsule of *C. neoformans* wild-type strain and various CAP mutants in vivo. (B) The quantification of capsule thickness in vivo for each strain. Each dot represents an individual fungal cell pooled from three mice per group. (C) Representative images showing the association of microglia with wild type and CAP mutants. CX3CR1$^{gfp/+}$ mice were i.v. infected with $1 \times 10^7$ TRITC-labeled *C. neoformans* for various time points. The brains were imaged by 20× objective after tissue clearing. (D) The percentage of *C. neoformans* with microglia association at 24 and 48h. CX3CR1$^{gfp/+}$ mice ($n = 3$ mice/group) were i.v. infected with $1 \times 10^7$ strains labeled with TRITC for different time points. (E) Schematic depiction of the *C. neoformans* capsule polysaccharides. Top, the structure of GXM. Bottom, the structure of GalXM. Mannose residues, green spheres; xylose residues, red stars; glucuronic acid residues, half-filled diamonds; galactose, yellow spheres. Single polymer repeat units are shown and all sugars are in the pyranose form unless labeled with f to indicate furanose. The key genes related to the synthesis of GXM and GalXM are highlighted in red. (F) Representative image showing the association of microglia with *uge1*Δ strain 24h post i.v. infection, with quantification shown on the right. (G) Representative image showing the association of microglia with *uxs1*Δ strain 24h post i.v. infection, with quantification shown on the right. The data underlying this Figure can be found in S1 Data. Scale bars: 20 μm (A) (F) (G), 50 μm (C). * $p < 0.05$, *** $p < 0.001$ by one-way ANOVA (D).

major mediator of microglial response. We next developed a convenient in vivo labeling strategy to quantify the proliferation of non-fluorescent fungi within individual fungal colonies (S6A Fig). Compared to wild type strain which rapidly proliferates (S6B Fig), the acapsular mutants (S6C–S6E Fig), as well as *uge1Δ*, *uxs1Δ* and *pka1Δ* (S6F–S6H Fig), all showed impaired proliferation in the brain. Collectively, these results suggested that GXM production is required for microglial response, and microglia-recruiting capability of *C. neoformans* does not guarantee in vivo proliferation.

## Microglia P2Y12 mediates recruitment towards capillary

Although chemokine receptor CX3CR1 is implicated in microglia recruitment [29,31], its deficiency did not impair microglia movement toward *C. neoformans* (S3D Fig), suggesting alternative mechanisms are involved. Microglia express multiple purinergic receptors, among which P2X7 and P2Y12 are the most predominant (S7A and S7B Fig). Previous studies have shown that extracellular nucleotides induce chemotaxis of cultured microglia through Gi/o-coupled P2Y receptors [36], and expression of P2Y12 is involved in microglial enwrapping of leaking blood vessel [37]. Since *C. neoformans* also induces vessel leakage [13], these previous studies inspired us to explore the role of purinergic receptors during microglia recruitment. We found pharmaceutical blockade of P2Y12 purinergic receptor by Clopidogrel significantly inhibited microglia recruitment but not with P2X7 inhibitor oATP (Fig 6A and 6B).

To rule out the possible side effects of P2Y12 inhibitors during microglia recruitment to *C. neoformans*, we obtained P2Y12$^{-/-}$ mice and bred them with CX3CR1$^{gfp}$ mice to label microglia. The successful knockout of P2Y12 was confirmed by flowcytometry (S7C Fig). Consistent with drug inhibition results, P2Y12$^{-/-}$ mice also showed delayed microglial response to *C. neoformans* (Fig 6C and 6D), which is associated with slightly reduced fungal burden in the brain (Fig 6E). The reduced fungal burden is only seen in the brain but not in other organs (Fig 6F), suggesting the growth-promotive role of microglia [15]. Thus, microglia purinergic receptor P2Y12 is involved in microglia recruitment towards *C. neoformans*.

## Fungal secreted GXM induces endothelial release of nucleotides

As P2Y12 senses both ADP and ATP [38], we next sought to address the source of nucleotides in our model. *C. neoformans* induces leakage of plasma, however, we found the concentration of ATP in plasma is low, unlikely sufficient to trigger microglia recruitment (S8A Fig). Activated platelets release ADP from dense granules upon blood vessel damage [39]. Thus, we questioned whether platelet is involved in microglial response. However, we did not observe substantial increase of platelets around fungal colonies (S8B and S8C Fig). Moreover, depletion of platelet had no effect in microglial response (S8D and S8E Fig).

ATP is typically sequestered within cells [40,41], and serves as a danger signal upon release [5,6,42]. Although damaged neurons can release ATP, neuron damage unlikely occurs before fungal BBB transmigration. Upon examining the interaction between *C. neoformans* and its surrounding environment, we found that *C. neoformans* intimately interacts with endothelial cells (Fig 7A). Although ATP can be released after cell death, endothelial cell death was not detected around *C. neoformans* even after prolonged time (Fig 7B). Notably, we found GXM released by *C. neoformans* was uptaken by endothelial cells (Fig 7C), suggesting it may trigger ATP release. To confirm this, we applied purified GXM to brain endothelial cultures for ATP release checking. Although ATP release was marginally increased after GXM treatment, the addition of ecto-ATPase inhibitor ARL [42] enhanced ATP detection, suggesting GXM induced endothelial ATP release is rapidly degraded by surface enzymes (Fig 7D). Additionally, we confirmed the increase in ATP was not attributable to cell death (Fig 7E). Together, microglial response to *C. neoformans* requires GXM-induced ATP release from endothelial cells, which activates microglial P2Y12 to induce migration.

## Discussion

In current study, we presented a paradigm for parenchyma-residing macrophages to detect pathogens without direct contact. In contrast to sensing microbial-released chemoattractants, *C. neoformans* recruits microglia indirectly through release of capsular GXM to acting on endothelial cells as illustrated in S9 Fig.

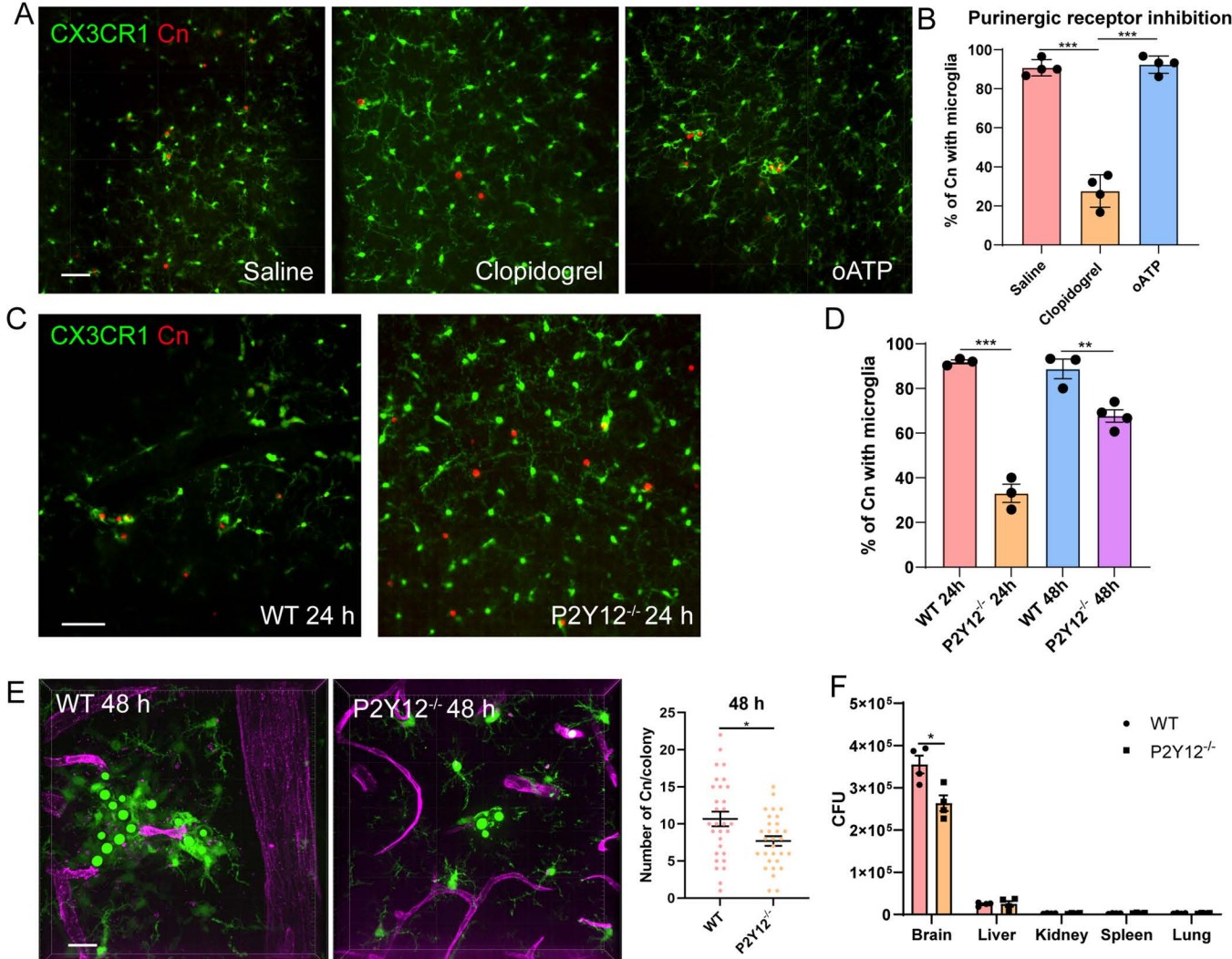

**Fig 6. Microglial P2Y12 mediates recruitment to *Cryptococcus neoformans*. (A)** Representative images showing the association of microglia with *C. neoformans* after treatment with the P2Y12 inhibitor Clopidogrel or the P2X7 inhibitor oATP. **(B)** Quantification of microglia association with *C. neoformans* 24 h after treatment with purinergic receptor inhibitors. **(C)** Representative images showing the association of microglia with *C. neoformans* in CX3CR1gfp/+ (WT) and CX3CR1gfp/+ P2Y12−/− mice. Mice were i.v. infected with 1 × 10^7 TRITC-labeled *C. neoformans* for 24 h. **(D)** The percentage of *C. neoformans* with microglia association in WT and P2Y12−/− mice. Mice (n = 4 mice/group) were i.v. infected with 1 × 10^7 TRITC-labeled *C. neoformans* for indicated time points. **(E)** Quantification of fungal proliferation 48 h after i.v. infection of CX3CR1gfp/+ (WT) and CX3CR1gfp/+ P2Y12−/− mice with 1 × 10^7 H99-GFP (n = 4 mice/group). **(F)** The quantification of CFU in different organs in CX3CR1gfp/+ (WT) and CX3CR1gfp/+ P2Y12−/− mice after infection with 5 × 10^4 H99 for 3 days. The data underlying this Figure can be found in S1 Data. Scale bars: 50 μm (A) (C), 20 μm (E). * p < 0.05, * p < 0.05, *** p < 0.001 by one-way ANOVA (B) (D), unpaired student *t* test (F).

Microglia have been shown to be recruited by many stimuli, including injured neurons [6], α-Syn aggregates [7], and blocked [8] or injured blood vessels [37]. We hereby added that demonstrating that rapid microglial recruitment can also be induced by intravascular pathogens, even before their parenchyma entry. Microglia can migrate rapidly towards leaking blood vessels to seal the leakage and prevent excessive inflammation [6,8,37]. Since *C. neoformans* is known to BBB leakage [43], it is reasonable to assume the primary purpose of microglia recruitment is likely to seal the leaking blood vessels, and microglia incidentally encountered *C. neoformans* hiding behind the capillaries.

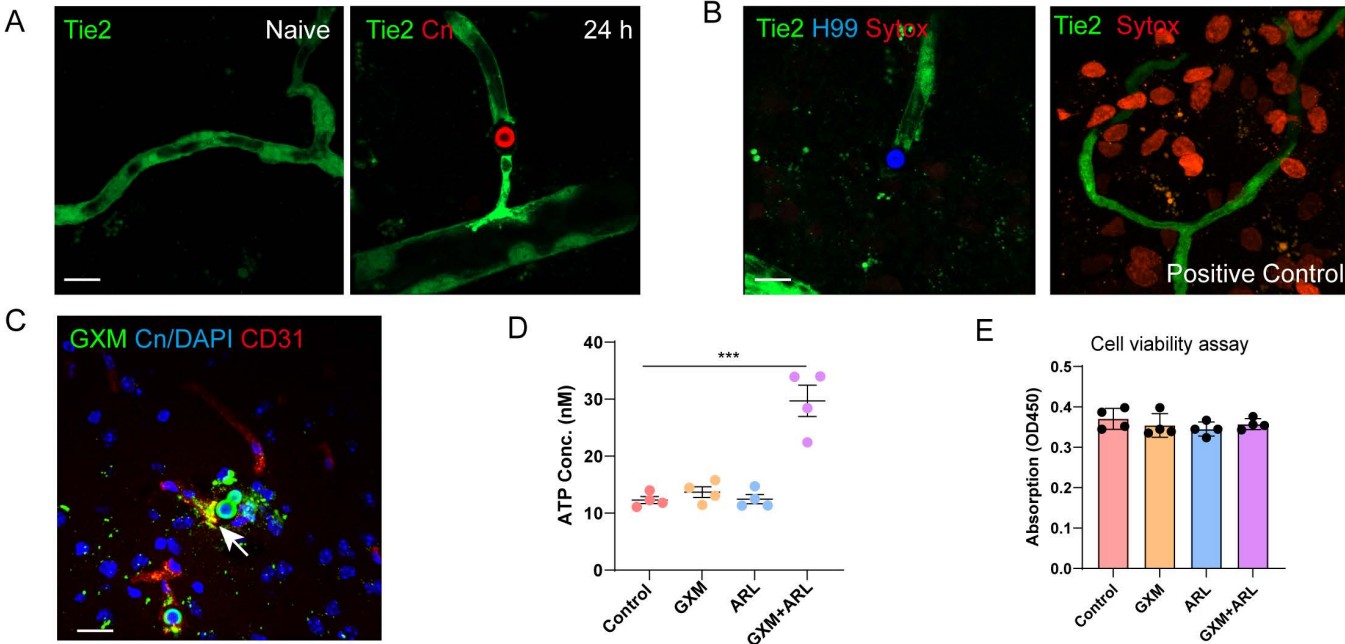

**Fig 7. Cryptococcal GXM stimulates endothelial cells to release ATP. (A)** Vascular endothelial cells in Tie2-GFP mice with and without *C. neoformans* (TRITC labeled) 24 h after infection. **(B)** Representative image showing the lack of Sytox Orange (an indicator of cell viability) staining on endothelial cells around *C. neoformans* (left), artificial injury induced by puncture with a 30G needle was included as positive control (right). **(C)** Representative immunofluorescence images showing the release of GXM by *C. neoformans* and its uptake by endothelial cells. **(D)** ATP release into the supernatant was measured in bEnd.3 endothelial cells treated with cryptococcal GXM (100 µg/mL), with or without 200 µM ecto-ATPase inhibitor ARL 67156 ($n = 4$). **(E)** The cell viability was measured for group settings as in (D) by CCK8 assay ($n = 4$). The data underlying this Figure can be found in S1 Data. Scale bar: 20 µm (A) (B) (C). *** $p < 0.001$ by one-way ANOVA (D).

Unfortunately, microglia association failed to inhibit *C. neoformans* under natural conditions, confirming previous study [15]. One possible explanation for the enhanced proliferation in microglia is the increased nutritional supply within macrophages, as *C. neoformans* has been shown to exploit host nutritional factors to promote growth [44]. We also noted microglia became metabolically active as shown by enlarged cell body and the accumulation of intracellular vesicles. Full activation of microglia may require the development of T cells [45], which normally takes at least two weeks [46]. Consistent with these studies, although microglia are rapidly recruited, we did not detect significantly increased microglia activation markers at the early stage. Notably, microglia are not required for BBB transmigration of *C. neoformans*, as we frequently observed fungal colonies without microglia association after microglia depletion. Previous reports suggest microglia can degrade BBB by phagocytosing endothelial cells in stroke model [8]. Although we did not detect apparent microglial uptake of endothelial cells, it is possible that microglia helped in degradation of extracellular matrixes to promote fungal brain invasion.

Microglia are highly dynamic under resting conditions [4] and are capable of migrating towards various stimuli. CX3CR1 has been implicated in microglia migration [29,31]. However, CX3CR1 is dispensable for microglia migration towards *C. neoformans*, possibly due to the lack of CX3CL1 induction by *C. neoformans*. Another study showed α-Syn is a chemoattractant for microglia via CD11b binding [7], however, since α-Syn is not induced in our model, CD11b is unlikely to be required. Our study aligns with previous models showing P2Y12 signaling is required for microglial response [5,37,38,47]. P2Y12, predominantly expressed by microglia, is a receptor for ADP and ATP. Membrane ectoenzymes can hydrolyze ATP into ADP and AMP, eventually to inosine. Whether these enzymes play roles in microglia recruitment to *C. neoformans* could be explored in further studies.

We showed fungal GXM induces endothelia ATP release and nucleotide receptor P2Y12 is involved in microglia recruitment. However, the exact role of ATP and how it acts on P2Y12 remain elusive. Previous studies have shown that ATP is not a chemoattractant itself, but can promote chemotaxis via autocrine/paracrine fashion [31,41,48]. Thus, we suggest that ATP is likely playing an amplification role during microglia recruitment towards *C. neoformans*. At this point, the exact chemokines mediating microglia recruitment also need further study. An in vitro study showed fibrinogen and albumin may be the chemokines to microglia [8]; however, since both factors are too essential to knockout, plus their receptors are not yet identified, we were unable to confirm their roles during *C. neoformans* induced microglia recruitment. In addition, whether ATP release by endothelial cells can trigger cytokines/chemokines that recruit microglia is still interesting to study.

Our lab has experiences of using intravital imaging to study host-pathogen interactions [27,28], however, we were not fortunate enough to capture the process of microglial migration towards *C. neoformans*. Unlike previous laser-induced injury models which triggers immediate damage and microglia recruitment, microglial response to *C. neoformans* is highly unpredictable for each single fungus. The limited field of view during intravital imaging only allows visualization of few fungal cells at a time. Thus, instead of capturing real-time movies, we employed 3D imaging of larger tissue areas. By exchanging space for time, we eventually analyzed sufficient events to support our conclusions.

Current study identified endothelial cells as the source of nucleotides. Nucleotides are ubiquitous extracellular modulators of cell functions [49], which activate G protein-coupled P2Y receptors and ionotropic P2X receptors on plasma membrane. Previous studies have shown that ATP induces rapid microglial response to local brain injury [6,38]. Various stimuli, such as mechanical stress [49], physical perturbation [49], and infection [5,42], can trigger ATP release from cells through various channels [49]. The involvement of these channels in microglia recruitment is of interest. In addition, necrotic and apoptotic cells can release ATP as a "find-me" signal [50], however, the lack of endothelial cell death around *C. neoformans* does not support ATP release from endothelial necroptosis or apoptosis. We detected very low levels of ATP in the plasma, although we could not completely rule out the possibility, but plasma is unlikely the major source of ATP detected by microglia in our model.

Our study also cast light on how *C. neoformans* transmigrates the BBB into the brain, which has been intensively studied over the past decades. Current knowledge suggest three major pathways, paracytosis, transcytosis, and Trojan horse pathways, which likely coexist during natural infection [43]. Monocytes are professional phagocytes which may transport fungal cells across BBB [28]. However, we showed monocytes have difficulties in phagocytosing fungal cells stuck in the capillaries, discouraging the role of monocyte during fungal transmigration. Our results also supported that *C. neoformans* primarily enter BBB from capillaries and that microglia played a role in aiding fungal proliferation after phagocytosis. Thus, microglia wrapping followed by endothelial retraction is likely a more efficient way of BBB transmigration by *C. neoformans*.

Compared to encapsulated *C. neoformans*, microglial response is not as robust in nonencapsulated fungal species, even they can physically penetrate BBB, such as *C. albicans*. These findings highlight the importance of the capsule in eliciting microglial responses and suggest that microglial responses triggered by capsular components is more efficient than mechanical puncturing of BBB. Supporting this concept, microglial responses are much reduced once *C. neoformans* has entered the parenchyma as shown in S3A and S3B Fig. Our data indicate that the capsule building blocks, rather than the assembled capsule, is required for microglial response. Notably, acapsular strains may still produce polysaccharide molecules [24], which explains why *cap64Δ* strain eventually triggered weak microglial response. We have several lines of evidence confirming capsular GXM, but not GalXM, induces microglia recruitment. The *cap59Δ* and *pka1Δ* strain, defective in GXM production but positive in GalXM [20,51], is not able to recruit microglia. On the other hand, the *uge1Δ* strain, defected in GalXM but positive in GXM, recruited equivalent microglia as wild type strain.

GXM is a complex, high-molecular-weight molecule that determines fungal serotypes [23]. While its structural motifs have been characterized, the biosynthesis, transportation and assembly of this capsular macromolecules remains unclear [23]. Moreover, the secondary structure of GXM is still unclear. Interestingly, the *uxs1Δ* strain, which lacks key xylose

residue on GXM, retains the ability to recruit microglia—consistent with presence of capsule production [26]. Whether other modifications, like glucuronic acid and O-acetylation of GXM, are required for microglial response remains to be determined. As capsular polysaccharides are synthesized intracellularly and secreted via vesicles [22], the pathways governing GXM export also warrant further investigation.

In conclusion, using the brain-invading pathogen *C. neoformans*, we revealed a novel mechanism by which microglia respond to pathogens without direct contact. Considering the rapid microglial response and the dramatic plasticity of macrophage, microglia re-educating strategies may be effective against *C. neoformans* brain infection.

## Materials and methods

### Ethics statement

All animal procedures were performed in accordance with institutional guidelines approved by the Institutional Animal Care and Use Committee (IACUC) of Shanghai Jiao Tong University (protocol number A2021045) and Shanghai Veterinary Research Institute Chinese Academy of Agricultural Sciences (protocol number SVLAC/XM-M-G23014).

### Animals

CX3CR1$^{gfp/gfp}$ mice (Strain #005582), originally from The Jackson Laboratory (Bar Harbor, ME, USA), were gifts from Prof. Jing Wang (Shanghai Jiao Tong University). CCR2$^{rfp/rfp}$ mice were gifts from Prof. Xiaoyu Hu (Tsinghua University). P2Y12$^{-/-}$ mice were gifts from Prof. Junling Liu (Shanghai Jiao Tong University). Tie2-GFP (Strain #003658), CX3CR$^{CreER/+}$ (Strain #021160), and ROSA26iDTR (iDTR mice, Strain #007900) mice were purchased from the Jackson Laboratory. Wild-type C57BL/6 mice were purchased from the Beijing Vital River Laboratory Animal Technology Co. The mice were cross-bred to generate CX3CR1$^{gfp/+}$ mice, CX3CR1$^{gfp/+}$CCR2$^{rfp/+}$, CX3CR1$^{gfp/+}$P2Y12$^{-/-}$, and CX3CR$^{CreER}$iDTR mice. For all experiments, male and female mice aged 6–12 weeks (weighing 18–22 g) were used. Animals were housed under a 12 h light/dark cycle at 22 ± 2 °C with ad libitum access to food and water.

### Fungal strains

*Cryptococcus neoformans* strains, including the high virulent serotype A strain *C. neoformans* var. grubii strain H99 (ATCC 208821) and serotype D strain B3501 (ATCC 34873) were purchased from ATCC and cryopreserved in glycerol stocks at −80 °C. Mutant strains (S1 Table) used in current study were from the *C. neoformans* var. grubii gene deletion collection purchased by L.W. from the Fungal Genetics Stock Center (FGSC, https://www.fgsc.net) or created in current study. All knockout strains were validated by PCR. In some experiments, the wild-type *Saccharomyces cerevisiae* S288c strain was a gift from Prof. Xinqing Zhao, and the *Candida albicans* SC5314 was a gift from Prof. Changbin Chen. For culturing, fungal strains were streaked from the glycerol stocks onto YPD agar plates (Cat#242720, BD) and incubated at 30 °C for 48 h to allow colony formation. Colonies were then inoculated into 15 mL of YPD broth and incubated overnight at 30 °C in an orbital shaker at 200 rpm. Fungal cells were harvested by centrifugation at 500$g$ for 5 min and washed twice with sterile phosphate-buffered saline (PBS). Fungal cell numbers were determined using a hemocytometer. In some experiments, fungi were heat-killed at 55 °C for 30 min. Only freshly harvested fungal cultures grown to log phase were used for all experiments to ensure consistency.

### Construction of fluorescent fungal strains

To generate fluorescent *C. neoformans* for in vivo tracking, a pTET-driven fluorescent protein cassette was integrated into the Safe Haven 2 locus of the fungal genome. The plasmid carrying pTET-driven tdTomato pYZ89 was kindly provided by Prof. Youbao Zhao. The tdTomato coding sequence was replaced with GFP using Gibson assembly. The resulting GFP or tdTomato expression cassettes, along with the hygromycin resistance gene, were fused to the 5′ and 3′ flanking regions (1.0–1.5 kb) of the Safe Haven locus via overlapping PCR. The insertion cassette with fluorescence protein and

resistance gene was introduced into the Safe Haven 2 locus of the recipient *C. neoformans* by the TRACE method. The plasmids encoding Cas9 and sgRNA were from Prof. Linqi Wang. Briefly, *C. neoformans* cells were cultured to log phase (OD600 = 0.6–1.0), harvested, and washed in ice-cold electroporation buffer (EB: 10 mM Tris-HCl pH 7.5, 1 mM $MgCl_2$, 270 mM sucrose, 1 mM DTT). After 1 h incubation on ice, cells were resuspended in 250 µL EB. A 45 µL aliquot was mixed with 5 µL DNA (1 µg Cas9, 700 ng sgRNA, 2 ng insertion cassette) and electroporated (2,000 V, 200 Ω, 25 µF, 4.5–5 ms). Cells were recovered in 1 mL YPD at 30 °C for 2 h, pelleted, and plated on selective YPD agar. Colonies were grown at 30 °C for 3 days. The brightest colonies were picked by a 10 µL tip for several generations to the obtain bright fluorescence. All fluorescent transformants were verified by PCR and tested for full virulence in mice.

## Chemical labeling of fungi

Chemical labeling is particularly useful for short-term tracking fungal cells, thus was frequently used in our i.v. infection models. For labeling fungi in vitro, the fungal surface was labeled with tetramethylrhodamine isothiocyanate (TRITC). Briefly, $1 \times 10^8$ cells were washed twice with PBS and resuspended in 1 mL PBS. TRITC (Sigma-Aldrich, Cat# 87918), prepared as a 10 mg/mL stock in DMSO, was added to a final concentration of 50 µg/mL. Cells were incubated at room temperature for 30 min in the dark with gentle rotation. TRITC labeling also allowed identification of mother cells within proliferating colonies derived from GFP-expressing H99 strains. Following incubation, cells were washed three times with PBS to remove excess dye, resuspended in PBS, and enumerated prior to infection. For in vivo labeling, mice were injected i.v. with 100 µL of 1% Uvitex 2B (Polysciences, Cat# 19517) in PBS 30 min before euthanasia. This method effectively labels *C. neoformans* with serum contact. All labeling procedures were performed under light-protected conditions to preserve fluorescence signal.

## Infection of mice

For the natural infection model, mice were i.n. infected with $1 \times 10^4$ fluorescently labeled fungal cells and monitored for 1-, 2-, or 3-weeks post infection. For i.v. infection model, mice were infected with $1 \times 10^7$ fungal cells through the tail vein and euthanized at 24 h or 48 h for imaging the fungal interactions with the host. In select experiments, mice were i.v. injected with $5 \times 10^4$ for evaluation of CFU 3 days post infection. Colony-forming units (CFUs) were then counted to quantify fungal load. To deliver *C. neoformans* directly into the brain parenchyma, intracerebral injections (i.c.) were performed using a stereotaxic apparatus (RWD Life Science). Mice were anesthetized with 250 mg/kg Avertin via intraperitoneal injection and secured on the stereotaxic frame. The scalp was disinfected and incised to expose the skull, and bregma was identified as a reference point. A small hole was drilled at defined coordinates (1.0 mm anterior, 1.5 mm lateral, and 2.0 mm deep from bregma for cortical injection). A Hamilton syringe fitted with a 30G needle was used to inject 5 µL of suspension at a rate of 1 µL/min. The needle was held in place for 5 min post-injection to prevent backflow, then slowly withdrawn. The incision was sutured, and mice were given analgesia and monitored on a warming pad until full recovery.

## Fungal burden enumeration

To determine organ fungal burden, infected mice at various time points were euthanized to harvest major organs. Each organ was placed into 15 mL tubes containing 2 mL of cold sterile PBS and homogenized using a handheld tissue homogenizer (TH220, Omni International). Serial dilutions of the tissue homogenates were plated onto YPD agar plates, and CFUs were counted after incubation at 30 °C for 36 h.

## Whole-mount confocal microscopy (in situ imaging)

To obtain in situ microscopic images of *C. neoformans* interacting with the host, whole-mount confocal microscopy of fixed brains was performed to minimize disturbance of the native spatial context. For visualizing the brain vasculature,

mice were i.v. injected with 50 µL of a 1 mg/mL DyLight 649-conjugated Tomato Lectin (Invitrogen, Cat# L32472) to label endothelial cells. Mice were anesthetized via intraperitoneal injection of 250 mg/kg Avertin working solution. Blood was collected by cardiopuncture, followed by transcardial perfusion with 20 mL of ice-cold PBS containing heparin (10 U/mL) to flush circulating blood cells, followed by 20 mL of 4% paraformaldehyde (PFA) for tissue fixation. Brains were carefully extracted and placed in 1 mL of PBS in a 35 mm glass-bottom imaging dish (#1.5 coverslip thickness; ibidi GmbH), with the dorsal surface facing downward for imaging on an inverted microscope. Images were obtained using Olympus IXplore SpinSR10 spinning disk confocal microscope equipped with a 60× oil immersion objective (NA = 1.30). Visible laser lines at 405, 488, 561, and 633 nm were employed for fluorescence excitation. This confocal system allowed imaging up to a depth of approximately 30 µm. For 3D imaging, z-stacks were acquired at 1 µm intervals over a total depth of 20–30 µm.

### Tissue clearing

For imaging larger areas of the brain, the collected brains were cleared by an organic-based tissue-clearing methods based on PEGASOS (polyethylene glycol-associated solvent system) which preserves endogenous fluorescence. Briefly, the method involves steps including fixation, delipidation, dehydration, and refraction index matching in clearing solution. mice were anesthetized and transcardially perfused with 20 ml ice-cold heparin PBS (10 U/mL heparin) followed by 20 mL 4% paraformaldehyde in PBS (pH = 7.4) by a peristaltic pump. Organs were post-fixed in 4% PFA at 4 °C overnight, then washed thoroughly with PBS. Samples were delipidated sequentially in 30%, 50%, and 70% tert-butanol (tB, Sigma-Aldrich, Cat# 360538), each supplemented with 3% Quadrol (Sigma-Aldrich, Cat# 122262) to adjust pH to above 9.5, for 12 h each step. Dehydration was achieved by incubating organs in tB-PEG dehydration solution composed of 70% v/v tB, 27% v/v PEGMMA500 (PEGMMA500, Sigma-Aldrich, Cat# 409529), 3% w/v Quadrol for 48 h at room temperature with gentle rotation. Tissues were subsequently transferred into the BB-PEG clearing solution, composed of benzyl benzoate (BB, Sigma-Aldrich B6630) and PEGMMA500 (7:3 v/v) with 3% Quadrol, for 24 h until fully transparent. Samples were stored and imaged in the clearing solution by Olympus IXplore SpinSR10 spinning disk confocal microscope equipped with Photometrics Prime 95B scientific CMOS camera. Although tissue clearing allows imaging of the entire brain using lower magnification objective; we used a 20× objective for 3D images (step size = 2 µm) to get high-quality images.

### Image analysis

The IMARIS 9.8 (Bitplane) software was used for visualization, measurement, 3D modeling, and movie generation. The size or distance was measured at Full Width at Half Maximum from a Gaussian fitted intensity profile. Volume-rendered stack images were created using the "volume rendering" function, optical slices were obtained with the "orthoslicer" tool. For 3D remodeling of structures, the "surface" function was used for each channel. In some analysis, the microglia cells were modeled using the "spot" function. For analysis of the microglia morphology, both the "surface" and "filament" function were used, allowing analysis of microglia volume as well as the total length and number of dendrites. 3D still images were captured via the "snapshot" function, and animations were produced using the "animation" module, movies were exported at a speed of 24 frame/s.

### Association of microglia with *C. neoformans*

Association of microglia with fungi in the brain was analyzed at the level of fungal colonies rather than individual fungal cells. A fungal colony was considered with microglia association if a microglial cell body was in direct contact (<2 µm) with any individual fungal cell within a fungal colony. For each mouse, at least 30 individual fungal colonies were analyzed throughout the brain cortex region or as many as possible for certain mice with fewer events.

## Monocyte depletion

For monocyte depletion, mice received i.v. injection of 200 µL Clodronate liposomes (Liposoma B.V.) 24 h before infection. Control mice received an equal volume of PBS liposomes. The efficiency of monocyte depletion was confirmed by flow cytometry of peripheral blood leukocytes.

## Immunofluorescence of brain slices

Mice were anesthetized by intraperitoneal injection of 250 mg/kg Avertin (2,2,2-tribromoethanol, Sigma-Aldrich, T48402), followed by sequential transcardial perfusion with PBS and 4% PFA. After post-fixation in 4% PFA over-night at 4 °C, brains were harvested and sequentially dehydrated in 15% and 30% sucrose solutions at 4 °C until the tissue sank. The brains were subsequently embedded in Tissue-Tek OCT Compound (Sakura Finetek, #4583), then snap-frozen in liquid nitrogen. Frozen brains were sectioned at a thickness of 20 µm using a cryostat microtome, and sections were mounted onto Superfrost Plus glass slides (Thermo Fisher Scientific). Slides were fixed in ice-cold acetone for 10 min, followed by blocking with 5% goat serum for 30 min at room temperature. Primary antibody incubation was performed overnight at 4 °C. The following primary antibodies were used: anti-Collagen IV (Chemicon, Cat# AB748), anti-GFP (Invitrogen, Cat# A-11122), anti-GXM (18B7, Sigma-Aldrich, Cat# MABF2069), CD31 (Biolegend, Cat# 102502), and anti-ZO-1 (Invitrogen, Cat# 33-9100). After three washes with PBS containing 0.05% Tween-20, sections were incubated with fluorophore-conjugated secondary antibodies (Invitrogen) for 2 h at room temperature. Following additional washes, nuclei were counterstained with DAPI, and slides were mounted using an antifade mounting medium for imaging by confocal microscope.

## Determination of endothelial cell viability in vivo

To determine endothelial cell viability in vivo, Sytox Orange (Invitrogen, Cat# S34861) was administered via i.v. injection at a dosage of 20 µg/mouse 10 min before euthanasia of Tie2-GFP mice previously infected with $1 \times 10^7$ C. neoformans. The brain was collected and visualized immediately using confocal microscope.

## Microglia depletion

Since microglia survival needs CSF1R signaling, mice were feed with diet containing 1,200 mg/kg PLX3397 (CSF1R inhibitor, Sunwaypharm, Shanghai, China) for 14 days to achieve microglia depletion. For microglia depletion in CX3CR1$^{CreER}$iDTR mice. Tamoxifen (Sigma-Aldrich, Cat# T5648) was given to mice via oral gavage as 500 µL of a 20 mg/mL solution in corn oil. Animals received two doses of 10 mg Tamoxifen given 48 h apart. Thirty days later (to allow full replacement of circulatory DTR positive cells), mice were i.p. treated with 1 µg/dose DT (Sigma-Aldrich, Cat# D0564) in 200 µL PBS for three consecutive days. For both treatments, the depletion of microglia was confirmed by fluorescence microscope or flowcytometry.

## Flowcytometry

Brains were harvested from euthanized mice following transcardial perfusion with 20 mL of cold PBS. Leukocytes were isolated using a 30% Percoll gradient centrifugation. To block Fc receptors, cells were incubated with anti-CD16/32 antibody (BioLegend, Cat# 101302) on ice for 10 min. Subsequently, cells were stained with fluorescently conjugated antibodies for 30 min, washed with staining buffer, and fixed with 1% paraformaldehyde in PBS. Samples were analyzed using a BD LSRFortessa II flow cytometer. The following antibodies were used: APC-Cy7 anti-CD45 (BioLegend, Cat# 103116), PE-Cy7 anti-CD11b (BioLegend, Cat# 101216), PE anti-CX3CR1 (BioLegend, Cat# 149006), PE anti-P2Y12 (BioLegend, Cat# 848004), BV421 anti-CD11c (BioLegend, Cat# 117343), BV510 anti-MHCii (BioLegend, Cat# 107636), PE anti-CD42d (BioLegend, Cat# 148504). Data analysis was performed using FlowJo software (version 10).

## Purinergic receptor blockade

To inhibit purinergic signaling, specific antagonists were used to block P2Y12 and P2X7 receptors. For P2Y12 inhibition, Clopidogrel (MedChemExpress LLC, HY-15283) was administered at a concentration of 50 mg/kg by oral gavage once daily. For P2X7 blockade, oxidized ATP (oATP, Sigma-Aldrich, 505758) was injected intraperitoneally at a dose of 30 mg/kg once daily. Both treatments were administered immediately after *C. neoformans* infection.

## GXM extraction

The GXM extraction methods were modified from previous studies. *C. neoformans* was cultured overnight in 1 L of YNB medium at 30 °C with shaking. Cells were pelleted, and the culture supernatant containing secreted polysaccharides was collected. Sodium acetate was added to a final concentration of 10% (w/v) with pH adjusted to 7.0 using acetic acid, followed by the addition of 2.5 volumes of ethanol to precipitate crude polysaccharides (GXM and GalXM). The mixture was left overnight, and the resulting precipitate was air-dried and dissolved in 2–3 mL of water. Total polysaccharide content was quantified using the phenol-sulfuric acid method. To purify GXM, the solution was adjusted to 0.2 M NaCl, and a 0.3% CTAB solution was slowly added (3:1 CTAB to polysaccharide, w/w) to selectively precipitate GXM, leaving GalXM in solution. The CTAB-GXM complex was pelleted by centrifugation and washed with 10% ethanol. To remove CTAB, the pellet was dissolved in 1 M NaCl, and GXM was re-precipitated by adding 2 volumes of ethanol. The GXM pellet was then dissolved in 2 M NaCl and dialyzed sequentially against 1 M NaCl overnight and distilled water for one week. The final GXM product was quantified again, lyophilized, and reconstituted in sterile PBS or culture medium for downstream applications.

## ATP detection

Mouse brain endothelial cell line bEnd.3 was cultured in 96 well plates until confluency. Cells were then treated with 100 µg/ml of purified capsular GXM for 1 h, with or without the addition of ecto-ATPase inhibitor ARL 67156 (200 µM, Sigma-Aldrich, Cat# A265) for 30 min, the cell culture supernatant was collected for detection of ATP release. ATP detection kit (Sigma-Aldrich, Cat# MAK190) was used for detection following manufacture's instruction. For the detection of ATP in plasma, mice with or without i.v. treatment with 25 U/mice apyrase (Sigma-Aldrich, Cat# A6535) were euthanized for blood collection 30 min after injection. Platelet-depleted plasma was obtained by centrifugation of freshly collected heparinized mouse blood at 2,000 x g for 15 min.

## Cell viability test

Cell viability was assessed using the Cell Counting Kit-8 (CCK-8, Dojindo) following the manufacturer's instructions. Cells were seeded in 96-well plates at a density of 5,000 cells per well, grown to confluency, and treated as indicated. After treatment, 10 µL of CCK-8 reagent was added to each well containing 100 µL of culture medium. Plates were incubated at 37 °C for 2 h, and absorbance was measured at 450 nm using a microplate reader. Cell viability was calculated after subtracting background absorbance from blank wells.

## Statistics

Data are presented as mean ± standard error of the mean. Statistical analyses were performed using GraphPad Prism 10 software (San Diego, CA). For comparisons between two groups, an unpaired two-tailed Student *t* test was used. For comparisons among more than two groups, one-way analysis of variance (ANOVA) followed by Tukey's post hoc test was applied to assess group differences. A *p*-value of <0.05 was considered statistically significant.

## Supporting information

**S1 Fig. Microglia respond rapidly to *Cryptococcus neoformans* in the brain. (A)** CX3CR1$^{gfp/+}$ mice were intranasally infected with $1 \times 10^4$ tdTomato-labeled *C. neoformans* H99 strain (H99-tdT) for 3 weeks. The brains were imaged by 4× objective after tissue clearing. **(B)** CX3CR1$^{gfp}$CCR2$^{rfp}$ dual reporter mice were i.n. infected with $1 \times 10^4$ H99-tdT for 3 weeks. Cleared brains were imaged using a 60× objective. Arrows indicate microglia interaction with fungal cells. **(C)** Tie2-GFP mice ($n = 3$ mice/group) were i.v. infected with $1 \times 10^7$ H99-tdT for 10 min or 24 h and imaged by whole-mount confocal imaging. Quantification of the percentage of intravascular fungi (right). **(D)** CX3CR1$^{gfp/+}$ mice were pretreated with 200 μL PBS liposome or Clodronate liposome for 12 h to deplete monocytes, followed by i.v. infection with $1 \times 10^7$ H99-tdT. Microglia recruitment was analyzed 24 h post-infection. **(E)** CX3CR1$^{gfp/+}$ mice were i.v. infected with $1 \times 10^7$ H99-tdT for indicated time points, the mean fluorescence intensity of microglia was measured for *C. neoformans* association with microglia (with the exception of naïve). Each dot represents an individual microglial cell pooled from three mice per group. **(F)** Representative 3D images as well as surface and filament rendering (branching and terminal points are displayed) showing the morphology of microglia (green) with or without association of *C. neoformans* (red). **(G)** The volumes of microglia with *C. neoformans* association at indicated time points. Each dot represents an individual microglia cell pooled from five mice per group. **(H)** The total dendrite length of microglia with *C. neoformans* association at indicated time points. Each dot represents an individual microglial cell pooled from five mice per group. **(I)** Representative image of microglia in the cerebellum (left). Total dendrite length of microglia in the cerebellum compared to those in the cortex (right). **(J)** Quantification of the percentage of *C. neoformans* with microglia association in the cerebellum. CX3CR1$^{gfp/+}$ mice ($n = 4$ mice/group) were i.v. infected with $1 \times 10^7$ H99-tdT for 24 h, the cerebellum region and the cortex region were imaged. **(K)** The survival curves of mice i.v. infected with $1 \times 10^7$ and $1 \times 10^5$ H99. The data underlying this Figure can be found in S1 Data. Scale bars: 700 μm (A), 20 μm (B) (C) (F) (I) (J), 50 μm (D) (E). *** $p < 0.001$ by one-way ANOVA (E) (G) (H).
(TIF)

**S2 Fig. Microglia recruitment depends on fungal viability but not proliferation. (A)** CX3CR1$^{gfp/+}$ mice were i.v. infected with $1 \times 10^5$ tdTomato-labeled *Cryptococcus neoformans* H99 strain (H99-tdT) for 48 h. The brain slices (1 mm thick) were imaged after tissue clearing, arrows indicate fungal location. **(B)** Representative image showing the recruitment of microglia to *C. neoformans* after low dose i.v. infection. Quantification of microglia association with *C. neoformans* after low dose i.v. infection was shown to the right ($n = 5$ mice). **(C)** CX3CR1$^{gfp/+}$ mice were i.v. infected with $1 \times 10^7$ TRITC-labeled B3501 strain for 24 and 48 h. Quantification of microglia association was shown to the right ($n = 3$ mice/group). **(D)** CX3CR1$^{gfp/+}$ mice were i.v. infected with $1 \times 10^7$ TRITC-labeled H99 strain for 24 h. The sizes of *C. neoformans* with or without microglia association were measured (right). Fungi freshly collected from rich medium were included as control. Each dot represents an individual fungus pooled from three mice per group. **(E)** CX3CR1$^{gfp/+}$ mice were i.v. infected with $1 \times 10^7$ TRITC-labeled heat-killed H99 by 24 and 48 h. Quantification of microglia association was shown to the right ($n = 4$ mice/group). **(F)** Representative images showing the relative location of beads with microglia after i.v. infection of $1 \times 10^7$ TRITC-labeled polystyrene beads of 8 μm for 24 and 48 h. Quantification of beads association with microglia at 24 and 48 h (right, $n = 3$ mice/group). **(G)** The growth of H99 and *ras1Δ* in YPD medium, confirming the temperature-sensitive property of *ras1Δ* strain. **(H)** Representative image showing the association of microglia with *ras1Δ* mutant strain after i.v. treated with $1 \times 10^7$ TRITC-labeled fungi for 24 h. Quantification of microglia association with *ras1Δ* fungi was shown to the right ($n = 3$ mice). **(I)** The analysis of microglial expression of activation makers CD11c and MHCii 2 days after i.v. infection with $1 \times 10^7$ H99. Refer to S7C Fig for gating strategy. The data underlying this Figure can be found in S1 Data. Scale bars: 20 μm (A) (B) (C) (D) (F). *** $p < 0.001$ by one-way ANOVA (B)(C).
(TIF)

**S3 Fig. Microglia interact with fungi emerging from the capillary vessel. (A)** A representative image showing the relative location of microglia after intracerebrally injected with $1 \times 10^3$ H99-tdT strain for 48 h (left). Quantification of the percentage of *Cryptococcus neoformans* with microglia association at indicated time points (right). CX3CR1$^{gfp/+}$ mice ($n = 3$ mice/group) were intracerebrally injected with $1 \times 10^3$ H99-tdT strain. **(B)** A representative image showing a microglia cell in association with a daughter cell in a colony with the mother cell already transmigrated the BBB (also see S5 Movie). **(C)** A representative imaging showing CX3CR1$^{gfp/gfp}$ mice also have microglia association with *C. neoformans* after i.v. infection with $1 \times 10^7$ H99-tdT. **(D)** Quantification of the percentage of *C. neoformans* with microglia association in CX3CR1$^{gfp/gfp}$ and CX3CR1$^{gfp/+}$ mice at indicated time points. Mice ($n = 4$ mice/group) were i.v. infected with $1 \times 10^7$ H99-tdT for 24 and 48 h. The brains were imaged by whole-mount confocal imaging using 60× objective for analysis of microglia association. **(E)** Flowcytometry identification of microglia in the brain. **(F)** The association of *C. neoformans* with microglia and other leukocytes in the brain. CX3CR1$^{gfp/gfp}$ and CX3CR1$^{gfp/+}$ mice ($n = 4$ mice/group) were i.v. infected with $1 \times 10^7$ H99-tdT for 24 h, the brain leukocytes were isolated and analyzed by flowcytometry. **(G)** The quantification of *C. neoformans* in association with microglia and other leukocyte were analyzed by flowcytometry. **(H)** A representative image showing GFP$^+$ monocyte in close proximity with *C. neoformans* after i.v. infection with $1 \times 10^7$ H99-tdT for 24 h. **(I)** Quantification of monocyte association with *C. neoformans* colonies as revealed by in situ imaging. **(J)** Statistics showing the percentage of *C. neoformans* being engulfed by monocytes as revealed by in situ imaging. CX3CR1$^{gfp/+}$ mice ($n = 4$ mice/group) were i.v. infected with $1 \times 10^7$ H99-tdT for 24 and 48 h. The data underlying this Figure can be found in S1 Data. Scale bars: 50 μm (A), 20 μm (B) (C) (H).
(TIF)

**S4 Fig. Acapsular *cap59Δ* strain has impaired ability to damage BBB. (A)** Immunofluorescence images showing the disruption of BBB by wild-type H99 strain as indicated by discontinuous collagen IV and CD31 staining (upper). In contrast, *cap59Δ* strain showed continuous collagen IV staining (lower). CX3CR1$^{gfp/+}$ mice were i.v. infected with $1 \times 10^7$ Uvitex 2B-labeled fungi for 24 h. **(B)** Immunofluorescence images showing the disruption of tight junction protein ZO-1 by wild-type H99 strain (left) and *cap59Δ* strain (right). Scale bar: 20 μm.
(TIF)

**S5 Fig. pka1Δ strain has reduced microglial association. (A)** Representative images showing the thickness of fungal strains in the brain. CX3CR1$^{gfp/+}$ mice were i.v. infected with $1 \times 10^7$ TRITC-labeled fungi for 24 h (left). Blood vessels were label by Alexa Fluor 649-conjugated Tomato Lectin (magenta). **(B)** Quantification of capsule thickness 24 h after i.v. infection. Each dot represents an individual fungal cell pooled from three mice per group. **(C)** Representative images showing the lack of microglia recruitment for *pka1Δ* strain, 20× (left), 60× (right). CX3CR1$^{gfp/+}$ mice were i.v. infected with $1 \times 10^7$ TRITC-labeled *pka1Δ* fungi for 48 h. **(D)** Quantification of microglia association with *pka1Δ* strain compared to wild-type H99 strain at indicated time points. **(E)** Representative images showing the lack of microglia recruitment and reduced capsule size of *pka1Δ* strain. **(F)** Quantification of capsule thickness of *pka1Δ* strain compared to H99. Each dot represents an individual fungal cell pooled from 3 mice per group. **(G)** Quantification of GXM release in culture medium of H99 and *pka1Δ* strain. The data underlying this Figure can be found in S1 Data. Scale bar: 10 μm (A), 20 μm (C right) (E), 50 μm (C left), * $p < 0.05$, *** $p < 0.001$ by one-way ANOVA (B)(D) or unpaired student *t* test (F) (G).
(TIF)

**S6 Fig. Proliferation of *Cryptococcus neoformans* in vivo quantified by Uvitex 2B staining. (A)** Representative image showing the successful labeling of *C. neoformans* in the brain by Uvitex 2B injection. Mice were i.v. injected with 100 μL 1% Uvitex 2B was injected through the tail vein 30 min before euthanasia to label *C. neoformans* in the brain. **(B)** The number of wild-type H99 cells per colony 24 h (left) or 48 h (right) post infection was enumerated. Mice were i.v. infected with $1 \times 10^7$ Uvitex labeled H99 and in vivo labeled by Uvitex 2B. **(C–H)** The number of fungal cells per colony for

indicated strains. Mice were i.v. infected with $1 \times 10^7$ Uvitex labeled fungi and analyzed 48 h post-infection. The data underlying this Figure can be found in S1 Data.
(TIF)

**S7 Fig. Microglia express high levels of P2Y12. (A)** The mRNA of P2X nucleotide receptors detected in microglia (classified as Hexb^hi and Klf^hi populations). Data obtained from Mouse Cell Atlas (MCA) website at http://bis.zju.edu.cn/MCA/. **(B)** The mRNA of P2Y nucleotide receptors detected in microglia. **(C)** Flowcytometry detection of P2Y12 in microglia and the confirmation of successful knockout in P2Y12$^{-/-}$ mice. The data underlying this Figure can be found in S1 Data.
(TIF)

**S8 Fig. Platelet is not required for microglia migration towards *Cryptococcus neoformans*. (A)** Quantification of ATP levels in the plasma of mice with or without i.v. injection of 25 U/mice apyrase 30 min before transcardial blood collection. **(B)** Representative images showing the successful labeling of platelet in naïve mice by PE-anti-CD42d antibody. **(C)** Representative images showing the distribution of platelets in around fungal clusters. CX3CR1$^{gfp/+}$ mice were i.v. infected with $1 \times 10^7$ H99-tdT for 24 h and 48 h, and platelet was labeled by PE-anti-CD42d antibody before euthanizing. **(D)** Representative images showing the recruitment of microglia to *C. neoformans* with or without platelet depletion by 200 µg/mice anti-CD41 antibody. CX3CR1$^{gfp/+}$ mice were i.v. infected with $1 \times 10^7$ H99-tdT for 24 h. **(E)** Quantification of microglia association with *C. neoformans* with or without platelet depletion 24 h post infection. The data underlying this Figure can be found in S1 Data. Scale bars: 20 µm (B) (C) (D).
(TIF)

**S9 Fig. The mechanistic illustration of current model.** (i) *Cryptococcus neoformans* trapped in capillaries release GXM. (ii) GXM induces endothelial activation and ATP release. (iii) Microglia P2Y12 detect ATP. (iv) Microglia migrate and enwrap capillary. (v) Microglia engulf *C. neoformans* and promote its growth.
(TIF)

**S1 Table. The information of mutant strains used in the study.**
(PDF)

**S2 Table. The association of microglia with mutant strains.**
(PDF)

**S1 Movie. Cryptococcus neoformans H99 brain infection induces robust recruitment of microglia** CX3CR1$^{gfp/+}$ mice were i.v. infected with $1 \times 10^7$ TRITC-labeled H99 strain for 24 h. The brain was cleared and imaged by whole-mount confocal imaging using 20× objective to obtain 3D images with a z step of 2 µm. A representative brain cortex region is shown demonstrating the robust association of microglia with fungal colonies. Green: GFP$^+$ microglia, Red: *C. neoformans* H99.
(MP4)

**S2 Movie. Microglia intimately interact with *Cryptococcus neoformans*.** CX3CR1$^{gfp/+}$ mice were i.v. infected with $1 \times 10^7$ H99-tdT strain for 24 h. The brains were cleared and imaged by whole-mount confocal imaging using 60× objective with a z step of 1 µm. A representative scene of microglia interaction with fungal colony is shown. Green: GFP$^+$ microglia, Red: *C. neoformans* H99 expressing tdTomato (H99-tdT).
(MP4)

**S3 Movie. The distribution of microglia and capillary vessels in the brain cortex region of naïve mice.** Following vasculature labeling by Tomato-Lectin, the brain of CX3CR1$^{gfp/+}$ mice was cleared and imaged by whole-mount confocal imaging using 20× objective to obtain 3D images with a z step of 2 µm. A representative region of brain cortex showing the

relative location of microglia and vasculature. Magenta: vasculature stained by DyLight 649-Tomato Lectin. Green: GFP$^+$ microglia in CX3CR1$^{gfp/+}$ mice.
(MP4)

**S4 Movie. Microglial association with daughter cell in a colony with transmigrated mother cell.** CX3CR1$^{gfp/+}$ mice were i.v. infected with $1 \times 10^7$ TRITC-labeled H99-GFP strain for 24 h. After labeling of vasculature by Tomato-Lectin, the brain was cleared and imaged by whole-mount confocal imaging using 60× objective to obtain 3D images with a z step of 1 μm. A representative scene of microglia interaction with the mother cell of a fungal colony is shown. Green: *Cryptococcus neoformans* and GFP$^+$ microglia in CX3CR1$^{gfp/+}$ mice, Red: Mother cell, Magenta: vasculature.
(MP4)

**S5 Movie. Microglia enwrapped capillary and eventually caused fungi phagocytosis.** CX3CR1$^{gfp/+}$ mice were i.v. infected with $1 \times 10^7$ TRITC-labeled H99 strain for 24 h. After labeling of vasculature by Tomato-Lectin, the brain was cleared and imaged by whole-mount confocal imaging using 60× objective to obtain 3D images with a z step of 1 μm. A representative scene of microglia wrapping up capillary vessels and eventually phagocytizing the fungi is shown. Green: GFP$^+$ microglia in CX3CR1$^{gfp/+}$ mice, Red: *Cryptococcus neoformans*, Magenta: vasculature.
(MP4)

**S6 Movie. Representative field of view showing reduced recruitment of microglia by *cap59Δ* strain 48 h post infection.** CX3CR1$^{gfp/+}$ mice were i.v. infected with $1 \times 10^7$ TRITC-labeled *cap59Δ* strain for 24 h. The brain was cleared and imaged by whole-mount confocal imaging using 20× objective to obtain 3D images with a z step of 2 μm. A representative brain cortex region is shown demonstrating the reduced association of microglia with *cap59Δ* strain. Green: GFP$^+$ microglia in CX3CR1$^{gfp/+}$ mice, Red: *Cryptococcus neoformans*.
(MP4)

**S7 Movie. Representative field of view showing reduced recruitment of microglia by *cap60Δ Cryptococcus neoformans* strains 48 h post infection.** CX3CR1$^{gfp/+}$ mice were i.v. infected with $1 \times 10^7$ TRITC-labeled *cap60Δ* strain for 24 h. The brain was cleared and imaged by whole-mount confocal imaging using 20× objective to obtain 3D images with a z step of 2 μm. A representative brain cortex region is shown demonstrating the reduced association of microglia with *cap60Δ* strain. Green: GFP$^+$ microglia in CX3CR1$^{gfp/+}$ mice, Red: *C. neoformans*.
(MP4)

**S8 Movie. Representative field of view showing reduced recruitment of microglia by *cap64Δ Cryptococcus neoformans* strains 48 h post infection.** CX3CR1$^{gfp/+}$ mice were i.v. infected with $1 \times 10^7$ TRITC-labeled *cap64Δ* strain for 24 h. The brain was cleared and imaged by whole-mount confocal imaging using 20× objective to obtain 3D images with a z step of 2 μm. A representative brain cortex region is shown demonstrating the reduced association of microglia with *cap64Δ* strain. Green: GFP$^+$ microglia in CX3CR1$^{gfp/+}$ mice, Red: *C. neoformans*.
(MP4)

**S9 Movie. Representative field of view showing substantial recruitment of microglia by *uge1Δ Cryptococcus neoformans* strains 48 h post infection.** CX3CR1$^{gfp/+}$ mice were i.v. infected with $1 \times 10^7$ TRITC-labeled *uge1Δ* strain for 24 h. The brain was cleared and imaged by whole-mount confocal imaging using 20× objective to obtain 3D images with a z step of 2 μm. A representative brain cortex region is shown demonstrating the substantial association of microglia with *uge1Δ* strain. Green: GFP$^+$ microglia in CX3CR1$^{gfp/+}$ mice, Red: *C. neoformans.*
(MP4)

**S10 Movie. Representative field of view showing reduced recruitment of microglia by *pka1Δ Cryptococcus neoformans* strains 48 h post infection.** CX3CR1$^{gfp/+}$ mice were i.v. infected with $1 \times 10^7$ TRITC-labeled *pka1Δ* strain for 24 h.

The brain was cleared and imaged by whole-mount confocal imaging using 20× objective to obtain 3D images with a z step of 2 µm. A representative brain cortex region is shown demonstrating the reduced association of microglia with *pka1Δ* strain. Green: GFP⁺ microglia in CX3CR1^gfp/+ mice, Red: *C. neoformans*.
(MP4)

**S1 Data. Numerical data behind all figures in Excel form.** The file contains multiple datasheets.
(XLSX)

## Acknowledgments

We thank the core facility of School of Life Sciences and Biotechnology, Shanghai Jiao Tong University for assistance with image acquisition and flowcytometry analysis.

## Author contributions

**Conceptualization:** Chenxu Feng, Ge Wang, Zongyan Chen, Donglei Sun.

**Formal analysis:** Ge Wang, Ziyi Ma, Donglei Sun.

**Funding acquisition:** Lisheng Zhang, Zongyan Chen, Donglei Sun.

**Investigation:** Chenxu Feng, Ge Wang, Yixuan Wang, Xiang Gao, Zhenqi Xu, Luyao Fang, Suwei Zheng, Yufeng Chu, Mei Meng, Weiwei Zhu, Lisheng Zhang, Donglei Sun.

**Methodology:** Chenxu Feng, Ge Wang, Yixuan Wang, Donglei Sun.

**Resources:** Yuyan Xie, Linqi Wang.

**Supervision:** Donglei Sun.

**Validation:** Chenxu Feng, Ge Wang.

**Visualization:** Ge Wang, Yixuan Wang, Donglei Sun.

**Writing – original draft:** Zongyan Chen, Donglei Sun.

**Writing – review & editing:** Chenxu Feng, Ge Wang, Angela Yang, Miriam Lu, Judd Denzel Garcia Mondina, Zongyan Chen.

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
