## [Editor Report · Decision Letter 0]

23 Sep 2025

Dear Dr Sun,

Thank you for submitting your manuscript entitled "Microglia monitor the capillaries for invading Cryptococcus neoformans before fungal transmigration of blood-brain barrier" for consideration as a Research Article by PLOS Biology.

Your manuscript has now been evaluated by the PLOS Biology editorial staff, as well as by an academic editor with relevant expertise, and I am writing to let you know that we would like to send your submission out for external peer review.

Once your full submission is complete, your paper will undergo a series of checks in preparation for peer review. After your manuscript has passed the checks it will be sent out for review. To provide the metadata for your submission, please Login to Editorial Manager (https://www.editorialmanager.com/pbiology) within two working days, i.e. by Sep 25 2025 11:59PM.

Kind regards,

Melissa

Melissa Vazquez Hernandez, Ph.D.

Associate Editor

PLOS Biology

---

## [Decision Letter · Decision Letter 1]

19 Nov 2025

Dear Donglei,

Thank you for your patience while your manuscript "Microglia monitor the capillaries for invading Cryptococcus neoformans before fungal transmigration of blood-brain barrier " was peer-reviewed at PLOS Biology. It has now been evaluated by the PLOS Biology editors, an Academic Editor with relevant expertise, and by three independent reviewers.

In light of the reviews, which you will find at the end of this email, we would like to invite you to revise the work to thoroughly address the reviewers' reports. As you will see below, majority of reviewers are positive about the relevance of the study, yet some concerns have raised during revision. Reviewer 1 questions the physiological relevance of the high inoculum used, noting it likely induces septic-like systemic inflammation, and requests survival or weight-loss data as well as experiments using lower-dose or chronic infection models to determine whether the microglial response is generalizable. The reviewer also highlights a discrepancy with prior literature, which reported microglial activation only by day 14, and call for an explanation supported by flow cytometry data and clarification of whether microglia distant from capillaries also activate. Reviewer 2 is fully supportive and raises no substantive concerns. Reviewer 3 asks to clarify the mechanistic link between GXM, endothelial ATP release, and microglial recruitment, to acknowledge the confounding systemic effects of PLX3397 depletion, and to strengthen the capsule-dependent conclusions by using acapsular C. neoformans controls rather than comparisons across species. We agree with all reviewer concerns and would require some additional experimental revisions to address them, as we consider that this would strengthen the work.

IMPORTANT: after discussion with the Academic Editor, we will ask you to address experimentally the requests of Reviewer 1, especially regarding the high dose. Therefore the experiments suggested with lower dose or the survival curves to provide context will be important for publication.

Given the extent of revision needed, we cannot make a decision about publication until we have seen the revised manuscript and your response to the reviewers' comments. Your revised manuscript is likely to be sent for further evaluation by all or a subset of the reviewers.

**IMPORTANT - SUBMITTING YOUR REVISION**

*Re-submission Checklist*

*Published Peer Review*

*PLOS Data Policy*

*Blot and Gel Data Policy*

Sincerely,

Melissa

Melissa Vazquez Hernandez, Ph.D.

Associate Editor

PLOS Biology

REVIEWERS' COMMENTS

Reviewer #1:

The authors report that brain-resident microglia vigilantly detect Cryptococcus neoformans within capillaries prior to its transmigration across the blood-brain barrier, but are less responsive to non-encapsulated fungi or parenchymal injections of C. neoformans. They further propose that the fungal capsule activates endothelial cells to release nucleotides, which then signal through microglial P2Y12. These are interesting and potentially important findings. However, I have several concerns outlined below.

Main Concerns

* The figures give the impression that too many microscopic images are crammed together, which sometimes distracts from the main narrative. The manuscript would benefit from streamlining by moving less essential images to supplementary figures and highlighting only the most important data in the main figures.

* Intravenous infection with C. neoformans H99 at 10^7 CFU is considered an "extremely" high inoculum (as supported by multiple prior publications). Such conditions represent a very acute and severe infection. Data on survival, or at least body weight loss, are needed to contextualize the infection severity used in these experiments. This is a great concern, as the mice may be experiencing sepsis-like systemic inflammation, which could trigger widespread responses throughout the body.

* Figures 1 and 2 suggest that microglia translocate toward capillaries within 24 hours of infection, with dendritic morphology changes observed in a similar timeframe. However, Neal et al. (PMID: 29162707) reported that microglia were not activated until day 14, based on CD11c and MHC-II flow cytometry. This discrepancy requires explanation. Is it a consequence of the unusually acute infection model? What about microglia located away from capillaries—do they also become recruited toward C. neoformans? A discussion, accompanied by flow cytometry data analyzing CD11c and MHC-II, would be important to address this.

* Related to the above, the very high inoculum likely induces a septic-like state with systemic inflammation. This represents a very specific condition that is not typical of most cryptococcal infections seen in clinics. The key question is whether a similar microglial response can be observed under chronic or lower-inoculum infection conditions. Data from lower-dose infections, where mice survive longer than one week, would be valuable.

* The manuscript should provide a clear diagram indicating precisely which brain regions were evaluated in the imaging analyses.

* Line 186 states that "microglia recruitment is strongly associated with vascular leakage." However, lectin staining delineates vascular structures and does not directly measure leakage. This needs clarification.

* The authors show that microglia recognize C. neoformans in capillaries. Do microglia also recognize the pathogen once it has entered the parenchyma, away from vessels? This point remains unclear.

Verbiage Issues

* The term "interaction" between microglia and C. neoformans may be overstated, since the data primarily show proximity rather than direct interaction. A more precise term should be used.

* The repeated use of "parenchyma-residing macrophages" is confusing, since the manuscript focuses on microglia. The terminology should be clarified.

* In Figure S3F and related analyses, the evaluation of fungus-host cell "proximity" based on "fungal localization" is unconventional. The authors should explain why this approach was chosen.

* Figure 2C: The y-axis is labeled "% of all C. neoformans with microglia," and one group is labeled "Within capillary." While the intent is clear (fungi within the capillary), the current wording could be misread as suggesting that microglia are inside capillaries. More precise labeling is needed.

* Figure 4F: The y-axis is also labeled "% of C. neoformans with microglia," but data from many groups presented are from peripheral organs. This labeling is confusing and should be revised.

Minor Points

* For Figure 1J, K, include representative images from each timepoint in the supplementary figures.

* Line 135: "Interestingly, although microglia in the cerebellum typically exhibit longer dendrites than those in the cortex (Fig. S1F), they are equally responsive to fungal invasion." The meaning of this sentence is not clear: Why are they "equally response" when they show longer dendrites?

* Figures 5B, I: Capsule thickness was inferred from spacing between stains. Why was capsule staining not performed directly with a GXM-specific antibody? Direct capsule labeling would strengthen the conclusions.

Reviewer #2:

This is an interesting manuscript on the role of microglial cells in cryptococcal CNS infection. This study reveals that brain-resident microglia can detect the fungus Cryptococcus neoformans before the yeast crosses the blood-brain barrier. Using advanced in situ imaging, the authors demonstrate that microglia rapidly migrate to and enwrap capillaries containing viable C. neoformans, an effect absent with nonencapsulated or dead fungi. Rather than killing the fungus, microglia uptake promotes fungal proliferation. The response is triggered not by direct fungal contact but by release of the Cryptococcus capsule polysaccharide glucuronoxylomannan (GXM) (i.e. CrAg), which activates endothelial cells to release nucleotides sensed by microglial P2Y12 receptors. Genetic and pharmacologic blockade of P2Y12 reduced microglial recruitment and fungal growth. These findings define a new paradigm of pathogen sensing by parenchymal macrophages—via endothelial signaling rather than direct contact. The authors suggest that modulating microglial activation may represent a therapeutic avenue against Cryptococcus. This last point is fanciful and impractical at present, based on the timing of when actual humans with cryptococcal meningitis present, but the authors are entitled to their opinion. The experiment of nature (i.e. reality), suggests that Th1 CD4 cells are critically important for control of Cryptococcus as absence of CD4s (e.g. HIV) is the global #1 risk factor.

Overall, this is well written, and I have no substantial comments.

Reviewer #3 (Rachael Dangarembizi):

This manuscript investigates the recruitment of microglia to C. neoformans trapped in brain capillaries and examines how microglia interact with the fungus before it crosses the blood-brain barrier. The study demonstrates that microglia detect pathogens indirectly via endothelial cell-derived danger signals triggered by fungal capsular components, particularly GXM. This represents a meaningful addition to our understanding of neuroimmune sensing and inflammatory signaling during cryptococcal infection. The manuscript is well written, and the authors present elegant spatial and temporal imaging data that illuminate a signalling axis in host-pathogen communication that has been largely unresolved in the field.

My main questions/concerns are:

1. The authors suggest a GXM-related recruitment of microglia which depends on the release of ATP by endothelial cells but the link between these events is not explained (Is ATP being released by damaged endothelial cells? The suggestion seems to be that the endothelial cells are intact. Alternatively, does ATP release by endothelial cells trigger cytokines/chemokines that recruit microglia? Or does ATP itself directly attract microglia? A causal link is necessary to helps readers understand the proposed signalling paradigm.

2. The emphasis of the decrease in CFUs and microglial association with C. neoformans after depletion should be moderated because a number of things change with the depletion including transmigration of fungi from the periphery (which depends on monocytes), altered physiology, signalling, cross talk in the remaining populations, one could argue that they are technically not in the same state as normal microglia. PLX3397's systemic and cellular effects should be acknowledged.

3. The claim that "nonencapsulated fungi fail to elicit robust microglial responses" is not fully supported, since the comparison is made across different fungal species with distinct morphologies, PAMP profiles, and pathogenic strategies (S. cerevisiae, C. albicans). A more appropriate comparison would be between encapsulated and nonencapsulated strains of C. neoformans, where the PAMP repertoire is conserved and capsule deficiency is the main variable. Incorporating or highlighting the data from acapsular C. neoformans mutants would strengthen the conclusion and avoid overgeneralisation.

---

## [Editor Report · Decision Letter 2]

9 Jan 2026

Dear Donglei,

I hope all is well. Thank you for your patience while we considered your revised manuscript "Microglia monitor the capillaries for invading Cryptococcus neoformans before fungal transmigration of blood-brain barrier" for publication as a Research Article at PLOS Biology. This revised version of your manuscript has been evaluated by the PLOS Biology editors and the Academic Editor.

Based on our Academic Editor's assessment of your revision, we are likely to accept this manuscript for publication, provided you satisfactorily address the remaining editorial points. Please also make sure to address the following data and other policy-related requests.

1) We routinely suggest changes to titles to ensure maximum accessibility for a broad, non-specialist readership, and to ensure they reflect the contents of the paper. In this case, we would suggest a minor edit to the title, as follows. Please ensure you change both the manuscript file and the online submission system, as they need to match for final acceptance:

"Microglia sense fungal infections through capsular components from capillary-bound Cryptococcus neoformans via endothelial nucleotide signaling"

2) The Ethics statement needs to be a separate, independent (and the first) subheading in the Material & Methods section. You currently have it within the subsection ."Animals and study protocols". Please place it before this. https://journals.plos.org/plosbiology/s/ethical-publishing-practice

Please supply the numerical values either in the a supplementary file or as a permanent DOI’d deposition for the following figures:

Figure 1DFH, 2BCH, 3BDG, 4BDEFH, 5DEFG, 6BDEF, 7DE, S1CEGHIJK, S2B-H, S3ADGIJ, S5BDFG, S6B-H, S7AB, S8AE

4) Please cite the location of the data clearly in all relevant main and supplementary Figure legends, e.g. “The data underlying this Figure can be found in S1 Data” or “The data underlying this Figure can be found in https://doi.org/10.5281/zenodo.XXXXX”

5) For figures containing FACS data (Figures 4G, S2I, S3EF, S7C), please provide the FCS files and a picture showing the successive plots and gates that were applied to the FCS files to generate the figure. We ask that you please deposit this data in the FlowRepository (https://flowrepository.org/) and provide the accession number/URL of the deposition in the Data Availability Statement in the online submission form.

6) Supplementary files (e.g., excel). Please ensure that all data files are uploaded as 'Supporting Information' and are invariably referred to (in the manuscript, figure legends, and the Description field when uploading your files) using the following format verbatim: S1 Data, S2 Data, etc. Multiple panels of a single or even several figures can be included as multiple sheets in one excel file that is saved using exactly the following convention: S1_Data.xlsx (using an underscore).

7) Please make sure that all figures use a colorblind-friendly palette.

8) Please note that per journal policy, the model system (mice) should be clearly stated in the abstract of your manuscript.

9) Please ensure that your Data Statement in the submission system accurately describes where your data can be found and is in final format, as it will be published as written there

10) Per journal policy, if you have generated any custom code during the course of this investigation, please make it available without restrictions. Please ensure that the code is sufficiently well documented and reusable, and that your Data Statement in the Editorial Manager submission system accurately describes where your code can be found. More information on our Code Policy, what and how to share can be found here: https://journals.plos.org/plosbiology/s/code-availability

We expect to receive your revised manuscript within two weeks.

*Published Peer Review History*

*Press*

Sincerely,

Melissa

Melissa Vazquez Hernandez, Ph.D.

Associate Editor

PLOS Biology

---

## [Editor Report · Decision Letter 3]

23 Jan 2026

Dear Donglei,

Thank you for the submission of your revised Research Article "Microglia sense fungal infections through capsular components from capillary-bound Cryptococcus neoformans via endothelial nucleotide signaling" for publication in PLOS Biology. On behalf of my colleagues and the Academic Editor, Rebecca Anne Drummond, I am pleased to say that we can in principle accept your manuscript for publication, provided you address any remaining formatting and reporting issues. These will be detailed in an email you should receive within 2-3 business days from our colleagues in the journal operations team; no action is required from you until then. Please note that we will not be able to formally accept your manuscript and schedule it for publication until you have completed any requested changes.

IMPORTANT: Many thanks for letting me know the issue with FlowRepository. Please upload the FCS files in a permanent repository like Zenodo and provide the DOI in the manuscript and Data Availability Statement. I have asked my colleagues to include this request alongside their own.

PRESS

Sincerely,

Melissa

Melissa Vazquez Hernandez, Ph.D., Ph.D.

Associate Editor

PLOS Biology
